# Comprehensive multiphase chlorine chemistry in the box model CAABA/MECCA: Implications to atmospheric oxidative capacity

Meghna Soni[1, 2], Rolf Sander[3], Lokesh K Sahu[1], Domenico Taraborrelli[4], Pengfei Liu[5], Ankit Patel[6], Imran A Girach[7], Andrea Pozzer[3, 8], Sachin S Gunthe[6, 9], and Narendra Ojha[1]

[1]Physical Research Laboratory, Ahmedabad, India
[2]Indian Institute of Technology, Gandhinagar, India
[3]Atmospheric Chemistry Department, Max Planck Institute for Chemistry, Mainz, Germany
[4]Institute of Energy and Climate Research, Troposphere (IEK-8), Forschungszentrum Jülich GmbH, Jülich, Germany
[5]School of Earth and Atmospheric Sciences, Georgia Institute of Technology, Atlanta, GA, USA
[6]EWRE Division, Department of Civil Engineering, Indian Institute of Technology Madras, Chennai, India
[7]Space Applications Centre, Indian Space Research Organisation, Ahmedabad, India
[8]Climate and Atmosphere Research Center, The Cyprus Institute, Nicosia, Cyprus
[9]Centre for Atmospheric and Climate Sciences, Indian Institute of Technology Madras, Chennai, India

**Correspondence:** Meghna Soni (soni.meghna95@gmail.com) and Rolf Sander (rolf.sander@mpic.de)

**Abstract.** Tropospheric chlorine chemistry can strongly impact the atmospheric oxidation capacity and composition, especially in urban environments. To account for these reactions, the gas- and aqueous-phase $Cl$ chemistry of the community atmospheric chemistry box model CAABA/MECCA has been extended. In particular, an explicit mechanism for $ClNO_2$ formation following $N_2O_5$ uptake to aerosols has been developed. The updated model has been applied to two urban environments with different concentrations of $NO_x$ ($NO + NO_2$): New Delhi (India) and Leicester (United Kingdom). The model shows a sharp build-up of $Cl$ at sunrise through $Cl_2$ photolysis in both the urban environments. Besides $Cl_2$ photolysis, $ClO+NO$ reaction, and photolysis of $ClNO_2$ and $ClONO$ are also prominent sources of $Cl$ in Leicester. High-$NO_x$ conditions in Delhi tend to suppress the night-time build-up of $N_2O_5$ due to titration of $O_3$ and thus lead to lower $ClNO_2$, in contrast to Leicester. Major loss of $ClNO_2$ is through its uptake on chloride, producing $Cl_2$, which consequently leads to the formation of $Cl$ through photolysis. The reactivities of $Cl$ and $OH$ are much higher in Delhi, however, the $Cl/OH$ reactivity ratio is up to $\approx 9$ times greater in Leicester. The contribution of $Cl$ to the atmospheric oxidation capacity is significant and even exceeds (by $\approx 2.9$ times) that of $OH$ during the morning hours in Leicester. Sensitivity simulations suggest that the additional consumption of VOCs due to active gas- and aqueous-phase chlorine chemistry enhances $OH$, $HO_2$, and $RO_2$ near sunrise. The simulation results of the updated model have important implications for future studies on atmospheric chemistry and urban air quality.

## 1 Introduction

Chlorine ($Cl$) radicals are one of the most important players in the tropospheric chemistry (Seinfeld and Pandis, 2016; Ravishankara, 2009). $Cl$ impacts the oxidative capacity of the atmosphere and radical cycling, and, therefore, can significantly alter the atmospheric composition (Seinfeld and Pandis, 2016; Faxon and Allen, 2013). In comparison with hydroxyl ($OH$) radicals,

the so-called atmospheric detergent, the much faster reaction rates of Cl with volatile organic compounds (VOCs), enhance the peroxy radicals ($RO_2$) formation and, thereby, the production of ozone ($O_3$) and secondary organic aerosols (SOA) (Qiu et al., 2019a; Choi et al., 2020). In addition, Cl radicals can also enhance the oxidation of climate-driving gases (such as methane and dimethyl sulphide) (Saiz-Lopez and von Glasow, 2012). Cl radicals are produced in the atmosphere through photochemistry involving heterogeneous reactions of Cl-containing gases and aerosols (Qiu et al., 2019a; Faxon and Allen, 2013). The major sources of Cl-containing species are anthropogenic activities in continental regions and sea salt aerosols in marine and coastal environments (von Glasow and Crutzen, 2007; Osthoff et al., 2008; Liao et al., 2014; Liu et al., 2017; Thornton et al., 2010; Gunthe et al., 2021; Zhang et al., 2022). The photolysis of reactive Cl-containing species, such as chlorine gas ($Cl_2$), hypochlorous acid (HOCl), nitryl chloride ($ClNO_2$), and chlorine nitrite (ClONO) and the reaction of hydrochloric acid (HCl) with OH are known to produce Cl radicals in the lower troposphere (Atkinson et al., 2007; Riedel et al., 2014). With the rise in anthropogenic activities, emissions of Cl-containing species have increased significantly across the globe (Lobert et al., 1999; Zhang et al., 2022), and hence the importance of Cl in local as well as regional atmospheric chemistry has become prominent.

Despite the aforementioned importance, Cl chemistry and associated mechanisms, especially heterogeneous reactions in the lower troposphere, are not yet fully understood, and the effects of Cl on atmospheric composition, air quality and oxidation capacity remain uncertain. Field measurements have revealed high concentrations of Cl species over inland regions in addition to coastal and polar regions (von Glasow and Crutzen, 2007; Osthoff et al., 2008; Liao et al., 2014; Liu et al., 2017; Thornton et al., 2010), however, quantitative understanding of continental sources remains poorly understood. This is due to lack of the relevant heterogeneous and gas-phase chemistry in atmospheric photochemical models despite the range of chemical mechanisms complexity used in 3-D chemistry transport models (Xue et al., 2015; Pawar et al., 2023; Pozzer et al., 2022). In addition, the chemistry of Cl compounds has been less studied using the laboratory/chamber experiments. Qiu et al. (2019b) showed that due to inadequate representation of heterogeneous Cl chemistry, the Community Multiscale Air Quality (CMAQ) model underestimated nitrate concentrations during daytime but overestimated during night-time in Beijing, China. In addition, the uncertainties associated with emission inventories of Cl species, can lead to inaccurate estimation of air composition (Zhang et al., 2022; Sharma et al., 2019). For example, Pawar et al. (2023) noticed that even after the inclusion of HCl emissions from trash burning the levels of nitrate, sulphate, nitrous acid (HONO) etc., still deviated from the observations in Delhi, India, highlighting the need to include emissions from other sectors, such as industries. A few recent studies assessed the impacts of the gas phase Cl chemistry by including gas phase $ClNO_2$ reactions, for example, Xue et al. (2015) reported about 25 % enhancement in the daytime oxidation of carbon monoxide and VOCs at a coastal site in East Asia. In the same region, the model predicted a 5-16 % enhancement in peak ozone with $ClNO_2$ ($\approx$50–200 pmol/mol) at a mountain top in Hong Kong, China (Wang et al., 2016). The measurements of $Cl_2$ (up to $\approx$450 pmol/mol) and $ClNO_2$ (up to $\approx$ 3.5 nmol/mol) were reported from a rural site in the North China Plain and Cl chemistry was showed to enhance the formation of peroxy radicals (by 15 %) and $O_3$ production rate (by 19 %) (Liu et al., 2017).

Nevertheless, the heterogeneous chemistry of Cl species remains poorly represented in models, and often neglected in large scale numerical simulations. For example, in several models, the heterogeneous uptake of $N_2O_5$ on aqueous aerosols yielded nitric acid ($HNO_3$) via reaction HET1:

$$N_2O_5(g) + H_2O(aq) \rightarrow 2\,HNO_3(aq) \qquad\qquad\qquad (HET1)$$

However, $N_2O_5$ uptake on chloride-containing particles can produce $ClNO_2$ (Behnke et al., 1997; Thornton et al., 2010) especially in urban environments with strong NOx emissions (Osthoff et al., 2008; Young et al., 2012). Incorporating heterogeneous mechanism of $ClNO_2$ into the regional models led to 3–12 % increase in $O_3$ over Northern China (Sarwar et al., 2014; Zhang et al., 2017; Liu et al., 2017). In addition, heterogeneous reactions of Cl-containing species including particulate chloride ($pCl^-$), $Cl_2$, $ClNO_2$, chlorine nitrate ($ClNO_3$), and hypochlorous acid (HOCl) are suggested to result in the formation of Cl radicals as well as in recycling of NOx, and HOx (OH, and $HO_2$) (Ravishankara, 2009; Qiu et al., 2019a; Hossaini et al., 2016; Faxon and Allen, 2013). Very recent measurements suggest a reduction in $ClNO_2$ formation due to the competition of $N_2O_5$ uptake among chloride, sulphate and acetate aerosols (Staudt et al., 2019). These heterogeneous reactions can be of paramount significance in the Cl budget, however, to the best of our knowledge, these are not yet considered in model simulations.

The main goal of the present study is to investigate the role of chlorine chemistry in chemically contrasting urban environments. In this regard, we incorporate comprehensive gas-phase and heterogeneous Cl chemistry into a state of the art box model. Section 2 provides a detailed description of the Cl chemistry mechanism with gas-phase and heterogeneous reactions. Section 3 describes the model setup and Section 4 shows the simulation results which include a detailed investigation on (i) the sensitivity of air composition to chlorine chemistry, (ii) the production and loss of Cl and $ClNO_2$, (iii) the role of Cl in the Atmospheric Oxidative Capacity (AOC), and (iv) the sensitivity to $ClNO_2 + Cl^-$ reaction.

## 2 Mechanism Development

The community box model "Chemistry As A Boxmodel Application/Module Efficiently Calculating the Chemistry of the Atmosphere" (CAABA/MECCA, Sander et al., 2019), has been used in this work. A comprehensive gas- and aqueous-phase mechanism of chlorine chemistry has been added to MECCA, here used within the box model CAABA. The gas-phase and heterogeneous chemistry implemented in MECCA is described in the following subsections and the full mechanism is shown in the supplementary section.

### 2.1 Gas-phase chlorine chemistry

A total of 36 inorganic, organic and photolysis reactions which are key contributors of Cl radicals were added to the mechanism (Table 1). The mechanism includes the inorganic reactions of Cl with NOx, $NO_3$ (G1–G4), the reactions of Cl-containing species with OH and NO (G5–G7), and the reactions between Cl-containing species (G8–G9) (Qiu et al., 2019a; Burkholder et al., 2015; Atkinson et al., 2007). ClONO is formed through reaction of Cl with $NO_2$ (G2), and exists as a metastable

intermediate (Janowski et al., 1977; Niki et al., 1978; Golden, 2007). This intermediate subsequently transforms into $ClNO_2$

(G10), with an average conversion time of $\approx$12 h (ranging from 4 to 20 h), and the corresponding rate constant is 2.3 E-5 s$^{-1}$ (Janowski et al., 1977). The Cl-initiated oxidation of organic species i.e. alkanes ($C_3H_8$, $C_4H_{10}$), aromatics (benzene ($C_6H_6$), toluene ($C_7H_8$) and xylene ($C_8H_{10}$)), alcohols ($CH_3OH$, $C_2H_5OH$), ketones ($CH_3COCH_3$, MEK), isoprene ($C_5H_8$), and other organic compounds ($C_2H_5CHO$, $HOCH_2CHO$, BENZAL, GLYOX, MGLYOX) have also been included (G11–G31). The corresponding kinetic data are based on the International Union of Pure and Applied Chemistry and NASA Jet

Propulsion Laboratory data evaluations (Atkinson et al., 2006, 2007; Burkholder et al., 2015), and from the literature (Niki et al., 1985, 1987; Green et al., 1990; Shi and Bernhard, 1997; Sokolov et al., 1999; Thiault et al., 2002; Wang et al., 2005; Rickard, 2009; Wennberg et al., 2018). In addition, photolysis reactions (G32–G36) resulting in production of Cl are also added to the module (Atkinson et al., 2007). The abbreviations of species mentioned in Table 1 are kept similar to that in the Master Chemical Mechanism (MCM) nomenclature (Rickard, 2009).

Table 1: Gas-phase chlorine reactions and corresponding rate constants added to MECCA. The rate constants are expressed in units of cm$^3$ molecule$^{-1}$ s$^{-1}$ unless otherwise specified. Model-simulated maximum noontime $J$-values for Delhi are provided.

| Reaction | | Rate constant | Reference |
|---|---|---|---|
| **Inorganic reactions** | | | |
| (G1) | $Cl + NO + M \rightarrow ClNO$ | 7.6E(-32)*(T/300)$^{-1.8}$ | Qiu et al. (2019a) |
| (G2) | $Cl + NO_2 + M \rightarrow ClONO$ | 1.6E-11 | Burkholder et al. (2015) |
| (G3) | $Cl + NO_2 + M \rightarrow ClNO_2$ | 3.6E-12 | Burkholder et al. (2015) |
| (G4) | $Cl + NO_3 \rightarrow ClO + NO_2$ | 2.40E-11 Qiu et al. (2019a) | |
| (G5) | $Cl_2 + OH \rightarrow HOCl + Cl$ | 3.6E-12*exp(-1200/T) | Atkinson et al. (2007) |
| (G6) | $ClNO_2 + OH \rightarrow HOCl + NO2$ | 2.4E-12*exp(-1250/T) | Atkinson et al. (2007) |
| (G7) | $OClO + NO \rightarrow NO_2 + ClO$ | 1.1E-13*exp(350/T) | Atkinson et al. (2007) |
| (G8) | $Cl + Cl_2O \rightarrow Cl_2 + ClO$ | 6.2E-11*exp(130/T) | Atkinson et al. (2007) |
| (G9) | $ClO + OClO + M \rightarrow Cl_2O_3$ | 1.2E-12 | Atkinson et al. (2007) |
| (G10) | $ClONO \rightarrow ClNO_2$ | 2.3E-5 s$^{-1}$ | Janowski et al. (1977) |
| **Organic reactions** | | | |
| (G11) | $Cl + C_3H_8 \rightarrow iso\text{-}C_3H_7O_2 + HCl$ | 1.4E-10*0.43*exp(75/T) | Rickard (2009) |
| (G12) | $Cl + C_3H_8 \rightarrow n\text{-}C_3H_7O_2 + HCl$ | 1.4E-10*0.59*exp(-90/T) | Rickard (2009) |
| (G13) | $Cl + iso\text{-}C_4H_{10} \rightarrow iso\text{-}C_4H_9O_2 + HCl$ | 1.43E-10*0.564 | Rickard (2009) |
| (G14) | $Cl + iso\text{-}C_4H_{10} \rightarrow tert\text{-}C_4H_9O_2 + HCl$ | 1.43E-10*0.436 | Rickard (2009) |
| (G15) | $Cl + n\text{-}C_4H_{10} \rightarrow LC_4H_9O_2 + HCl$ | 2.05E-10 | Atkinson et al. (2006), Rickard (2009) |
| (G16) | $Cl + benzene \rightarrow C_6H_5O_2 + HCl$ | 1.3E-16 | Sokolov et al. (1999) |

| | | | |
|---|---|---|---|
| *(G17)* | Cl + toluene $\rightarrow C_6H_5CH_2O_2$ + HCl | 6.20E-11 | Wang et al. (2005) |
| *(G18)* | Cl + isoprene $\rightarrow$ .63 LISOPAB + .30 LISOPCD + .07 LISOPEFO2 + HCl | 7.6E-11*exp(500/T)*1.1*exp(-595/T) | Wennberg et al. (2018) |
| *(G19)* | Cl + isoprene $\rightarrow$ .63 LISOPAB + .30 LISOPCD + .07 LISOPEFO2 + LCHLORINE | 7.6E-11*exp(500/T)*(1-1.1*exp(-595/T)) | Wennberg et al. (2018) |
| *(G20)* | Cl + xylene $\rightarrow C_6H_5CH_2O_2$ + LCARBON + HCl | 1.50E-10 | Shi and Bernhard (1997) |
| *(G21)* | Cl + $CH_3OH \rightarrow HOCH_2O_2$ + HCl | 7.1E-11*0.59*exp(-75/T) | Atkinson et al. (2006) |
| *(G22)* | Cl + $C_2H_5OH \rightarrow HOCH_2CH_2O_2$ + HCl | 6.0E-11*exp(155/T)*0.28*exp(-350/T) | Atkinson et al. (2006) |
| *(G23)* | Cl + $C_2H_5OH \rightarrow C_2H_5O_2$ + HCl | 6.0E-11*exp(155/T)*(1-0.28*exp(-350/T)) | Atkinson et al. (2006) |
| *(G24)* | Cl + $HOCH_2CHO \rightarrow$ HOCHCHO + HCl | 8.0E-12/0.9*0.35 | Atkinson et al. (2006), Niki et al. (1987) |
| *(G25)* | Cl + $HOCH_2CHO \rightarrow HOCH_2CO$ + HCl | 8.0E-12/0.9*(1-0.35) | Atkinson et al. (2006), Niki et al. (1987) |
| *(G26)* | Cl + GLYOX $\rightarrow$ HCOCO + HCl | 3.8E-11 | Niki et al. (1985) |
| *(G27)* | Cl + MGLYOX $\rightarrow CH_3CO$ + CO + HCl | 4.8E-11 | Green et al. (1990) |
| *(G28)* | Cl + $C_2H_5CHO \rightarrow C_2H_5CO_3$ + HCl | 1.3E-10 | Atkinson et al. (2006) |
| *(G29)* | Cl + $CH_3COCH_3 \rightarrow CH_3COCH_2O_2$ + HCl | 1.5E-11*exp(-590/T) | Atkinson et al. (2006) |
| *(G30)* | Cl + MEK $\rightarrow LMEKO_2$ + HCl | 3.05E-11*exp(80/T) | Atkinson et al. (2006) |
| *(G31)* | Cl + BENZAL $\rightarrow C_6H_5CO_3$ + HCl | 1.0E-10 | Thiault et al. (2002) |

| **Photolysis reactions** | | $J$-value (s$^{-1}$) | |
|---|---|---|---|
| *(G32)* | ClO $\rightarrow$ Cl + O3P | 1.45E-4 | Atkinson et al. (2007) |
| *(G33)* | $Cl_2O \rightarrow$ Cl + ClO | 9.20E-4 | Atkinson et al. (2007) |
| *(G34)* | $Cl_2O_3 \rightarrow$ ClO + $ClO_2$ | 5.50E-4 | Atkinson et al. (2007) |
| *(G35)* | ClNO $\rightarrow$ Cl + NO | 2.89E-3 | Atkinson et al. (2007) |
| *(G36)* | ClONO $\rightarrow$ Cl + $NO_2$ | 3.81E-3 | Atkinson et al. (2007) |

## 95   2.2   Heterogeneous chemistry

The aqueous-phase and heterogeneous chemistry of Cl compounds added to the MECCA is described in Table 2. In the present study, we assume that $N_2O_5$ is in equilibrium between the gas- and aqueous-phase (H2) according to Henry's law and the dissociation of $N_2O_5(aq)$ to nitronium ion ($NO_2^+$) and nitrate ($NO_3^-$), occurs according to reaction (A1). The rate constant for the recombination reaction of $NO_2^+$ and $NO_3^-$ is $2.7 \times 10^8$ mol$^{-1}$ L s$^{-1}$, calculated based on Bertram and Thornton (2009);

Staudt et al. (2019). The acid dissociation of nitric acid ($HNO_3$) in aqueous phase (A3) also results in formation of $NO_2^+$ (Sapoli et al., 1985).

Table 2: Aqueous-phase and heterogeneous chlorine reactions added to MECCA

| Reaction | | Rate constant | Reference |
|---|---|---|---|
| **Aqueous-phase reactions** | | | |
| (A1) | $N_2O_5(aq) \rightarrow NO_2^+(aq) + NO_3^-(aq)$ | $1.5 \times 10^5$ s$^{-1}$ | Staudt et al. (2019) |
| (A2) | $NO_2^+(aq) + NO_3^-(aq) \rightarrow N_2O_5(aq)$ | $2.7 \times 10^8$ mol$^{-1}$ L s$^{-1}$ | Bertram and Thornton (2009); Staudt et al. (2019) |
| (A3) | $HNO_3(aq) + H^+(aq) \rightarrow NO_2^+(aq) + H_2O(aq)$ | $1.6 \times 10^9$ mol$^{-1}$ L s$^{-1}$ | Sapoli et al. (1985) |
| (A4) | $NO_2^+(aq) + Cl^-(aq) \rightarrow ClNO_2(aq)$ | $7.5 \times 10^9$ mol$^{-1}$ L s$^{-1}$ | Staudt et al. (2019) |
| (A5) | $ClNO_2(aq) \rightarrow NO_2^+(aq) + Cl^-(aq)$ | $2.70 \times 10^2$ s$^{-1}$ | Behnke et al. (1997) |
| (A6) | $ClNO_2(aq) + Cl^-(aq) \rightarrow Cl_2(aq) + NO_2^-(aq)$ | $10^7$ mol$^{-1}$ L s$^{-1}$ | Roberts et al. (2008) |
| (A7) | $OH \cdot Cl^-(aq) + OH \cdot Cl^-(aq) \rightarrow Cl_2(aq) + 2\ OH^-(aq)$ | $1.8 \times 10^9$ mol$^{-1}$ L s$^{-1}$ | Knipping et al. (2000) |
| (A8) | $OH \cdot Cl^-(aq) + Cl^-(aq) \rightarrow Cl_2^-(aq) + 2\ OH^-(aq)$ | $10^4$ mol$^{-1}$ L s$^{-1}$ | Grigorev et al. (1987) |
| (A9) | $Cl_2^-(aq) + 2\ OH^-(aq) \rightarrow OH \cdot Cl^-(aq) + Cl^-(aq)$ | $4.5 \times 10^7$ mol$^{-1}$ L s$^{-1}$ | Grigorev et al. (1987) |
| (A10) | $NO_2^+(aq) + H_2O(aq) \rightarrow HNO_3(aq) + H^+(aq)$ | $1.6 \times 10^7$ mol$^{-1}$ L s$^{-1}$ | Staudt et al. (2019) |
| (A11) | $NO_2^+(aq) + SO_4^{2-}(aq) \rightarrow SO_4^{2-}(aq) + NO_3^-(aq) + 2\ H^+(aq)$ | $7.5 \times 10^9$ mol$^{-1}$ L s$^{-1}$ | Staudt et al. (2019) |
| (A12) | $NO_2^+(aq) + HCOO^-(aq) \rightarrow HCOO^-(aq) + NO_3^-(aq) + 2\ H^+(aq)$ | $7.5 \times 10^9$ mol$^{-1}$ L s$^{-1}$ | Staudt et al. (2019) |
| (A13) | $NO_2^+(aq) + CH_3COO^-(aq) \rightarrow CH_3COO^-(aq) + NO_3^-(aq) + 2\ H^+(aq)$ | $7.5 \times 10^9$ mol$^{-1}$ L s$^{-1}$ | Staudt et al. (2019) |
| (A14) | $NO_2^+(aq) + phenol(aq) \rightarrow HOC_6H_4NO_2(aq) + H^+(aq)$ | $7.5 \times 10^9$ mol$^{-1}$ L s$^{-1}$ | Ryder et al. (2015); Heal et al. (2007) |
| (A15) | $NO_2^+(aq) + CH_3OH(aq) \rightarrow CH_3NO_3(aq) + H^+(aq)$ | $4.5 \times 10^8$ mol$^{-1}$ L s$^{-1}$ | Iraci et al. (2007) |
| (A16) | $NO_2^+(aq) + CRESOL(aq) \rightarrow TOL1OHNO2(aq) + H^+(aq)$ | $7.5 \times 10^9$ mol$^{-1}$ L s$^{-1}$ | Coombes et al. (1979) |

| **Heterogeneous reactions** | | **Henry's constant (mol L$^{-1}$ atm$^{-1}$)** | |
|---|---|---|---|
| (H2) | $N_2O_5(g) \rightleftharpoons N_2O_5(aq)$ | 8.8E-2 | Fried et al. (1994) |
| (H3) | $ClNO_2(g) \rightleftharpoons ClNO_2(aq)$ | 4.5E-2 | Frenzel et al. (1998) |

| (H4) | $HOC_6H_4NO_2(aq) \rightleftharpoons HOC_6H_4NO_2(g)$ | 8.9E1 | Müller and Heal (2001) |
|---|---|---|---|
| (H5) | $CH_3NO_3(aq) \rightleftharpoons CH_3NO_3(g)$ | 2.0E0 | Sander (2015) |

Thus produced nitronium ion ($NO_2^+$) reacts reversibly with chloride ($Cl^-$) yielding $ClNO_2$ (A4, A5) (Staudt et al., 2019; Behnke et al., 1997). After outgassing according to Henry's law (H3), $ClNO_2$ is photolyzed in the gas phase, producing Cl and $NO_2$ (Ghosh et al., 2012; Sander et al., 2014). $ClNO_2$ uptake on chloride containing aerosols results in formation of $Cl_2$ and nitrite ion ($NO_2^-$), as shown by the reaction (A6) (Roberts et al., 2008). Chamber experiments suggest the formation of $Cl_2$ from the self reaction of OH·$Cl^-$ (A7), which gets formed via the reaction of OH with $Cl^-$ (Knipping et al., 2000). Through other channel of reversible reactions (A8, A9), $OH \cdot Cl^-$ reacts with aqueous chloride and produces $Cl_2^-$, which can yield $Cl_2$ through subsequent reactions (Grigorev et al., 1987). The $NO_2^+$ uptake on aqueous chloride to form $ClNO_2$ (A4) is ≈500 times faster than $NO_2^+$ reaction with $H_2O$ (A10) (Staudt et al., 2019). At the same time, experimental studies revealed a strong competition of $NO_2^+$ to react with $Cl^-$ and with other nucleophiles (e.g. $SO_4^{2-}$) and aqueous organic compounds e.g. phenol, methanol, cresol (A11–A16) (Staudt et al., 2019; Ryder et al., 2015; Heal et al., 2007; Iraci et al., 2007; Coombes et al., 1979). These reactions could suppress the formation of $ClNO_2$ and also the corresponding rate constants for reactions A11–A14 are similar to the $NO_2^+$ + $Cl^-$ reaction yielding $ClNO_2$ i.e. $7.5 \times 10^9$ mol$^{-1}$ L s$^{-1}$ (Staudt et al., 2019; Ryder et al., 2015; Heal et al., 2007). Methanol reacts with $NO_2^+$ (A15) and forms aqueous methyl nitrate ($CH_3NO_3$) (Iraci et al., 2007). Phase exchange for $CH_3NO_3$ and nitrophenol ($HOC_6H_4NO_2$) is shown by reactions H4 and H5, respectively. The heterogeneous chemistry just discussed is implemented in MECCA and is summarized in Fig. 1. The rate constant for $NO_2^+$ reaction with methoxyphenol is about ≈10000 times smaller than $NO_2^+$ + phenol reaction (Kroflič et al., 2015), so it is not considered in this study. In addition, nitration reactions of other alcohols (e.g. catechol and polyphenols) could be potentially important, however due to unavailability of corresponding rate constants, these reactions are not considered in this study, nonetheless future studies calculating the kinetics of these reactions are recommended.

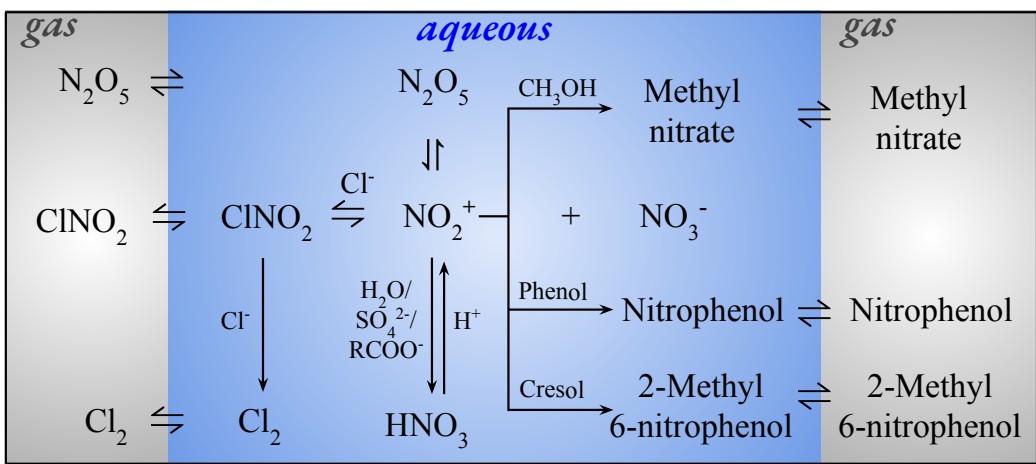

**Figure 1.** Aqueous-phase and heterogeneous chemistry added to MECCA.

In this study, Cl reacts with hydrocarbons and acetone via H-abstraction, and hence does not lead to the formation of any Cl-containing molecules, such as chloroacetone. This means that there are no such reactions in MECCA in which the Cl atom becomes part of the organic molecule. The reaction of Cl atoms with isoprene proceeds mainly via addition, and it produces chlorine-containing organics (Ragains and Finlayson-Pitts, 1997; Fan and Zhang, 2004). However, here we have simplified the mechanism by not considering the fate of organohalogens. Therefore, for future research, it would be valuable to investigate the chemical kinetics of such reactions kinetics and their importance in the formation of organohalogen compounds.

## 3  Box model setup

The chemistry described in section 2 has been added into community box model CAABA/MECCA v4.4.2 (Sander et al., 2019). A comprehensive gas and aqueous phase tropospheric chemistry involving total 3330 reactions was utilized for the simulations, and the full set of reactions are presented in the electronic supplement. The gas-phase chemistry of organics like terpenes and aromatics is treated by the Mainz Organic Mechanism (MOM) (Taraborrelli et al., 2012; Nölscher et al., 2014; Hens et al., 2014; Taraborrelli et al., 2021). The aqueous-phase chemistry of oxygenated VOCs is treated by the Jülich Atmospheric Mechanism of Organic Chemistry (JAMOC) (Rosanka et al., 2021). The numerical integration of the chemical mechanism is performed by the kinetic preprocessor v2.1 (KPP) (Sandu and Sander, 2006). The photolysis rate constants ($J$ values) are calculated by the submodel JVAL, based on the method by Landgraf and Crutzen (1998). The Cl chemistry is expected to be more prominent during winter conditions due to higher concentration of Cl-containing species in the boundary layer (Thornton et al., 2010; Gunthe et al., 2021; Sommariva et al., 2021), and therefore, simulations are performed for the winter season. Hence, the model is set-up for typical winter conditions of two different urban environments: Delhi (28.6° N, 77.2° E), India and Leicester (52.4° N, 01.1° W), United Kingdom. Simulations are performed for a 5-day period (17–21 February 2018) and output of $5^{th}$ day has been considered for the analysis; by then, radicals had achieved almost a steady state. The typical environmental conditions used in the simulations for Delhi (Tripathi et al., 2022) and Leicester (Sommariva et al., 2021) are summarized in Tab. 3 and Tab. S1.

VOC emissions are taken from the CAMS inventory (Sindelarova et al., 2014; Granier et al., 2019) and are adjusted iteratively in magnitude for better agreement with observations. CAMS-GLOB-ANT v5.3 (0.1° × 0.1°) (Granier et al., 2019) provides emissions of anthropogenic VOCs (e.g., benzene, toluene etc.), while emissions of biogenic VOCs (e.g., isoprene) are from CAMS-GLOB-BIO v3.1 (0.25° × 0.25°) (Sindelarova et al., 2014). Emission of HCl and particulate chloride are included from Zhang et al. (2022) and adjusted iteratively towards reported levels of Cl-containing species (Gunthe et al., 2021; Sommariva et al., 2021). The Mainz Organic Mechanism (MOM) dry deposition scenario (Sander et al., 2019) is activated in the model. Ground-based lidar measurements of boundary layer height (BLH) during winter-time, performed as a part of the European Integrated project on Aerosol Cloud Climate and Air Quality Interactions (EUCAARI) project, are utilized for the simulations at Delhi (Nakoudi et al., 2018). The diurnal variation in BLH in Leicester is extracted from the European centre for medium-range weather forecast's (ECMWF) fifth-generation reanalysis dataset ERA5 (Hersbach et al., 2020). Air compo-

**Table 3.** Environmental conditions of Delhi and Leicester in the model simulations.

| Parameter | Delhi | Leicester |
|---|---|---|
| Latitude | 28.58° N | 52.38° N |
| Longitude | 77.22° E | 01.08° W |
| Time-zone | GMT+5:30 | GMT+0:00 |
| Temperature (K) | 292 | 278.1 |
| Pressure (mbar) | 1010 | 1004 |
| Air number density (molecules $cm^{-3}$) | $2.5 \times 10^{19}$ | $2.61 \times 10^{19}$ |
| Relative Humidity | 67 % | 90 % |

sition in the model has been initialized based on previous studies (Tab. S1; Zhang et al. (2007); Lanz et al. (2010); Lawler et al. (2011); Sommariva et al. (2018, 2021); Gunthe et al. (2021); Tripathi et al. (2022)). The values of aerosol properties (e.g., radius, liquid water content, and chemical composition) incorporated in the simulations for both Delhi and Leicester as provided in Table. S1. We constrained the model with the parameterized function best representing the observed diurnal variations of NOx (Fig. 2) (Tripathi et al. (2022); Sommariva et al. (2018, 2021), https://uk-air.defra.gov.uk/data/) which helped in better reproducing the diurnal variations of some VOCs (e.g. isoprene) and ozone. Diurnal observations of HONO from Sommariva et al. (2021) are used for Leicester. For Delhi, however, HONO couldn't be constrained due to lack of observations.

## 4 Results and Discussion

The model captures the patterns in $O_3$ variability at both locations (Sommariva et al., 2018; Nelson et al., 2021; Chen et al., 2021; Sommariva et al., 2021; Nelson et al., 2023) to an extent, as shown in Fig. 2. $O_3$ is underestimated after $\approx$16:00 h LT in Leicester mainly due to titration by high NO and lack of adequate dynamics/transport of $O_3$ in the model. Entrainment seems to improve $O_3$ after mid-night, towards the observed values (Fig. 2i). Simulated isoprene is in agreement with diurnal observations in Delhi (Tripathi et al., 2022) and in accordance with observed mean level in Leicester (Sommariva et al., 2021). The nitrate radical ($NO_3$), which is a nighttime oxidant, is formed through reaction between $NO_2$ and $O_3$ (G37). $NO_3$ can react with $NO_2$ forming $N_2O_5$, which can again produce $NO_3$ and $NO_2$ through thermal dissociation (G38).

$$NO_2 + O_3 \longrightarrow NO_3 + O_2 \tag{G37}$$

$$NO_3 + NO_2 + M \rightleftharpoons N_2O_5 + M \tag{G38}$$

As seen in Fig. 2e, $NO_3$ remains negligible during the night-time ($\approx$18:00–07:30 h LT) in Delhi due to unavailability of $O_3$ under high-NO conditions (up to 200 nmol/mol). Interestingly, despite its very short lifetime ($\approx$5 s), about $\approx$0.1 pmol/mol of $NO_3$ sustains during daytime. This is primarily due to prevailing levels of $NO_2$ ($\approx$30 nmol/mol) and $O_3$ ($\approx$40 nmol/mol).

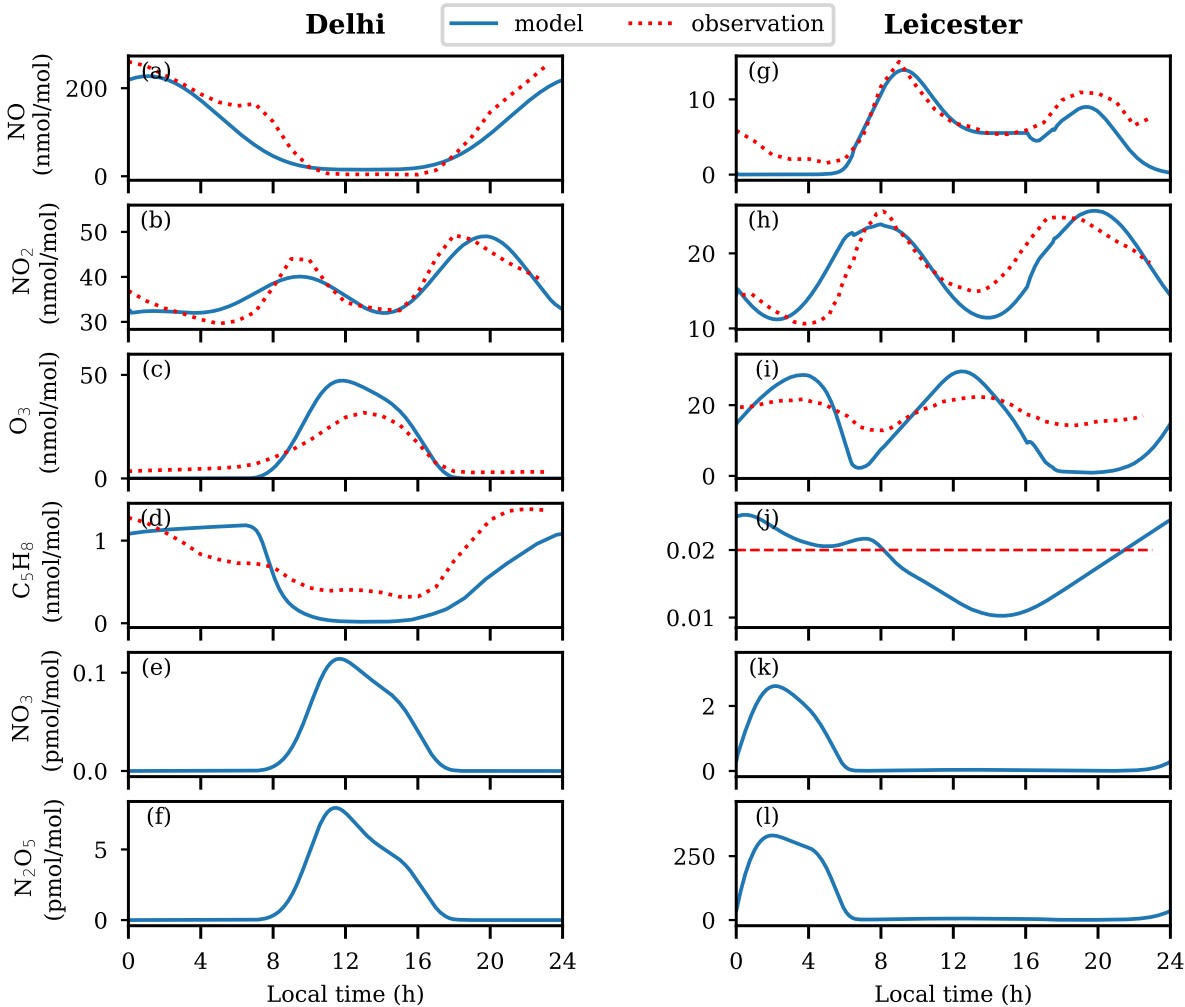

**Figure 2.** Diurnal variations of $NO, NO_2, O_3, C_5H_8, NO_3$, and $N_2O_5$ mixing ratios in Delhi (left) and Leicester (right). The unusal and negligible nighttime $NO_3$ in Delhi is attributed to the nearly non-existent $O_3$, due to titration by higher concentrations of NO. This leads to the negligible nighttime $N_2O_5$ in this region. Although mixing ratios of $NO_3$ and $N_2O_5$ peak during the daytime, their levels remain quite low. Mean value of observed $C_5H_8$ in Leicester is shown by red colored long dashed line.

Such unusual daytime enhanced $NO_3$ have been reported in recent studies, for example, 5-31 pmol/mol of $NO_3$ in Texas, USA (Geyer et al., 2003). Aircraft measurements during the New England Air Quality Study showed $\approx$0.5 pmol/mol of $NO_3$ within boundary layer ($\leq$1 km) during noon time (Brown et al., 2005). The calculated $NO_3$ levels using steady state approximation showed 0.01-0.06 pmol/mol of $NO_3$ for the 1997-2012 period at urban sites in the UK (Marylebone Road London, London Eltham, and Harwell) (Khan et al., 2015a). Horowitz et al. (2007) suggested that $NO_3$ in tenths of pmol/mol during daytime over the eastern United States results in formation of $\approx$50 % isoprene nitrates through oxidation of isoprene, which could

further affect the formation of $O_3$ and SOA significantly (Horowitz et al., 2007). Following to higher $NO_3$, up to 8 pmol/mol of $N_2O_5$ is simulated during daytime in Delhi (Fig. 2f). Similar unusual daytime high levels of $N_2O_5$ ($\approx$21.9$\pm$29.3 pptv) during wintertime were recently measured at Delhi using a high-resolution iodide adduct chemical ionization mass spectrometer (Haslett et al., 2023).

Enhanced $NO_3 \approx 2.6$ pmol/mol and $N_2O_5 \approx 330$ pmol/mol are simulated after mid-night in Leicester (Fig. 2k, 2l). In contrast to Delhi, the daytime simulated levels of $NO_3$ are negligible as it gets removed rapidly during the daytime by photolysis and through its reactions with NO, $HO_2$, $RO_2$, and VOCs (Khan et al., 2015b). In conjunction with high NO from $\approx$16:00 h LT to near midnight that titrates $O_3$, the corresponding $NO_3$ and $N_2O_5$ is negligible (following reactions G37 and G38). Night-time high and negligible day-time levels of $NO_3$ and $N_2O_5$ are their typical features which are generally reported in the literature (Brown et al., 2001; Seinfeld and Pandis, 2016).

## 4.1 Sensitivity of air composition to chlorine chemistry

To investigate the effects of Cl chemistry on air composition, other than comprehensive chemistry simulation discussed in previous section (simulation: NEW i.e. chemistry already present in the model + newly added gas and aqueous phase chlorine chemistry), two additional simulations have been performed, which are: (1) OLD – this includes default chemistry already present in the model, and (2) NOCL – OLD minus chlorine chemistry (i.e. without Cl chemistry). OLD simulation also encompassed some basic chlorine chemistry that was part of the model prior to its update (full mechanism is also shown in supplement). Figure 3 shows the comparison of Cl, $ClNO_2$, ClONO, OH, $HO_2$, and $RO_2$ variations among the three simulations in Delhi and Leicester. Figure S5 shows the differences in diurnal variations of Cl, ClONO + $ClNO_2$, OH, $HO_2$, and $RO_2$ in NEW simulation with: NOCL and OLD simulations.

The Delhi environment is mainly characterized by two peaks in Cl, a predominant sharp peak just after sunrise followed by a broad shallow peak during noontime, corresponding to different mechanisms as discussed in the next section. With newly added chemistry (NEW simulation), a sharp peak in Cl is seen near sunrise, with the maximum values attained is $\approx$3.5 fmol/mol ($8.75 \times 10^4$ molec cm$^{-3}$) in Delhi (Fig. 3a). A broad smaller peak with magnitude of $\approx$0.8 fmol/mol maximizing around noontime is seen, which is $\approx$4 times smaller than the first morning peak. OLD simulation also show a sharp peak in Cl near sunrise in Delhi, with a maximum of $\approx$11 fmol/mol ($2.75 \times 10^5$ molec cm$^{-3}$). Cl get suppressed by up to $\approx 0.01$ pmol/mol of maximum value in the OLD simulation, in the presence of added chlorine chemistry (NEW) as shown in Fig. S5. Similar to Cl, a peak is seen in ClONO + $ClNO_2$ of $\approx$100 pmol/mol with sunrise, which gradually decreases and attain $\approx$7 pmol/mol from nearly 11:00–16:00 h LT. Afterwards it increases to $\approx$20 pmol/mol from late evening as shown by Fig. 3b,c. The pathways for the formation of $ClNO_2$ and ClONO were absent in earlier version of the model (OLD). Simulated OH, $HO_2$, and $RO_2$ show a prominent peak just after sunrise in the presence of Cl chemistry for both the OLD and NEW simulations (Fig. 3d,e,f). As a consequence of greater oxidation of VOCs by Cl, enhanced levels of OH by 0.05 pmol/mol (up to a factor of $\approx$1.8), $HO_2$

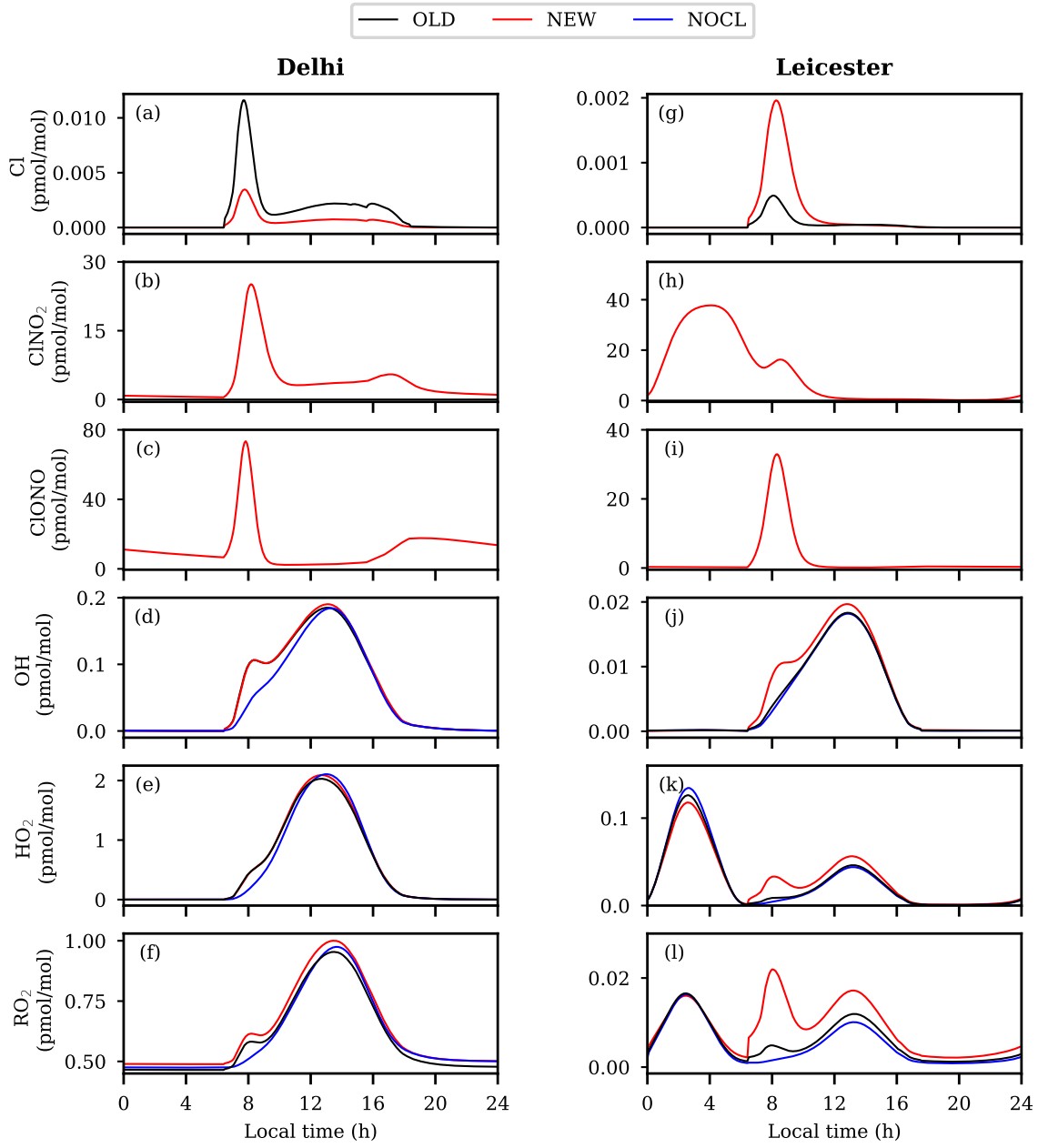

**Figure 3.** Model simulated diurnal variations of Cl, $ClNO_2$, ClONO, OH, $HO_2$, and $RO_2$ at Delhi (left panel) and Leicester (right panel).

by 0.21 pmol/mol and $RO_2$ by 0.1 pmol/mol are noted with added Cl chemistry compared to NOCL case (see Fig. S5). No significant changes are seen in noon-time levels of OH and $HO_2$, whereas $\approx 1.1$ times more $RO_2$ is produced with added Cl

chemistry (NEW) compared to the OLD simulation.

The model-predicted Cl peaks at $\approx$2 fmol/mol ($5.2 \times 10^4$ molec cm$^{-3}$) during sunrise in Leicester (Fig. 3g). In contrast to Delhi, suppressed Cl (up to $\approx$ 3.2 times) with a narrow peak is simulated by OLD simulation in comparison with NEW simulation containing newly added Cl chemistry, at Leicester. In contrast to negligible night-time ClONO + ClNO$_2$ in Delhi, it shows a strong build-up over Leicester during 0-4 hours with a maximum of $\approx$40 pmol/mol, with higher levels (up to

50 pmol/mol) prevailing until about sunrise. ClONO + ClNO$_2$ is negligible during mid-day until mid-night, in accordance with N$_2$O$_5$ in Leicester as shown in Fig. 2l. Previous studies have demonstrated that the formation of ClNO$_2$ occurs within the nocturnal residual layer, which contains lower levels of NO compared to the surface layer. Subsequently, ClNO$_2$ mixes downward during the morning when the convective mixed layer develops (Bannan et al., 2015; Tham et al., 2016). However, the present study does not account the the effect of transport processes due to the limitations of the box model. The effects

of added Cl chemistry on OH, HO$_2$, and RO$_2$ are more prominent in Leicester compared to Delhi. NEW simulation show strong enhancements in OH (up to $\approx$ 2 times), HO$_2$ (up to $\approx$ 5 times), and RO$_2$ (up to $\approx$ 8 times) after sunrise which is gradually progressive, resulting in higher levels during noon-time as well (Fig. 3, Fig. S5). Remarkably elevated levels of RO$_2$ (by a factor of $\approx$ 2) are prominent during the noon hours. Such elevated levels of RO$_2$ could favour enhanced levels of secondary organic aerosols in Leicester. The impact of Cl chemistry on aerosols (NO$_2^+$, NO$_3^-$, and oxalic acid) is discussed

in Supplementary section 2.2 (Fig. S6). Though significant differences in NO$_2^+$, NO$_3^-$, and oxalic acid are seen due to Cl chemistry, further measurements are required for validation. In the next sections, we have analysed the observed behaviour of Cl and ClNO$_2$ in the NEW simulation over both the locations in more detail.

## 4.2  Production and loss of Cl and ClNO$_2$

The sources and sinks of Cl in Leicester and Delhi are presented in Fig. 4. The left-upper panel (a) delineates the sources

and sinks of Cl radical on diurnal scale in Delhi. The morning sharp peak in Cl radical is caused mainly by the photolysis of Cl$_2$ with a maximum rate of 1.2 x 10$^7$ molec cm$^{-3}$ s$^{-1}$. The shallow secondary peak is due to the reaction HCl + OH with a noon time rate of $\approx$0.4 x 10$^7$ molec cm$^{-3}$ s$^{-1}$. However, there is a smaller contribution from other reactions (photolysis of ClNO$_2$, ClONO and reaction of ClO with NO) to the morning peak, which have negligible contributions during the daytime. Interestingly, there is a strong consumption of Cl to oxidize VOCs (peak rate $\approx$2.4 x 10$^7$ molec cm$^{-3}$ s$^{-1}$) during sunrise, and

a lesser consumption during the rest of the day. Cl + NO$_2$ is also a Cl sink during the morning time in Delhi. The Cl-initiated oxidation of VOCs in the morning hours in Delhi may lead to formation of secondary organic aerosols and new particle formation, which opens up pathways of future research in this direction. In addition to Cl$_2$ photolysis ($\approx$1.0 x 10$^6$ molec cm$^{-3}$ s$^{-1}$), photolysis of ClNO$_2$ and ClONO, and ClO + NO reaction (total rate $\approx$0.8 x 10$^6$ molec cm$^{-3}$ s$^{-1}$) are other prominent sources of Cl in Leicester. VOCs are the major sink for Cl (rate $\approx$1.3 x 10$^6$ molec cm$^{-3}$ s$^{-1}$), followed by NO$_2$ (rate $\approx$0.6 x

10$^6$ molec cm$^{-3}$ s$^{-1}$).

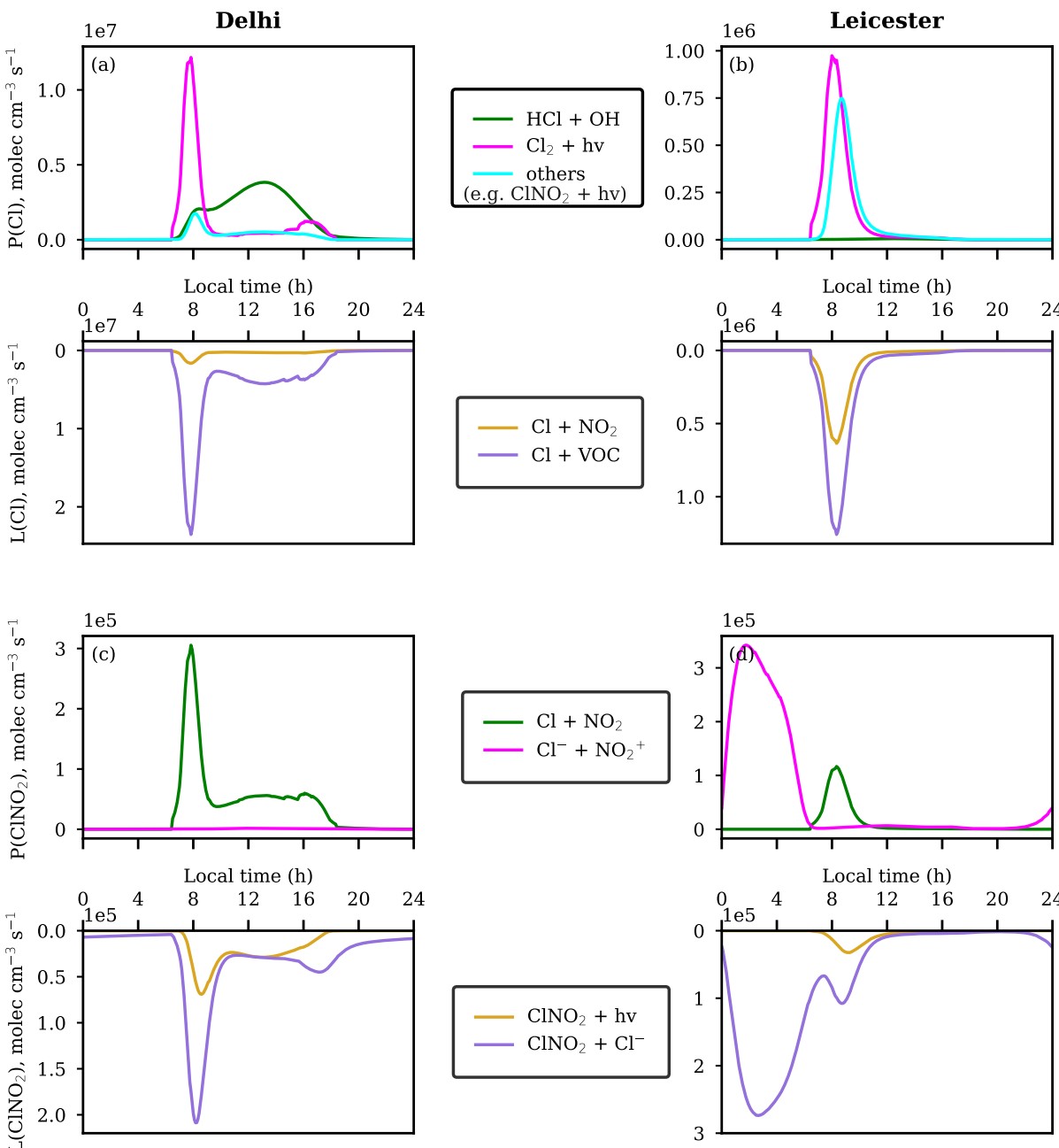

**Figure 4.** Production and loss rates of (a, b) $Cl$ and (c, d) $ClNO_2$ in Delhi (left panel) and Leicester (right panel).

We further analyzed the production and loss pathways of $ClNO_2$, as shown in Fig. 4c,d. While the major source of $ClNO_2$ is through the $Cl + NO_2$ reaction with a reaction rate $\approx 3 \times 10^5$ molec cm$^{-3}$ s$^{-1}$ in Delhi, the aqueous phase reaction $Cl^- + NO_2^+$

($\approx 3.4 \times 10^5$ molec cm$^{-3}$ s$^{-1}$) is the prominent source in Leicester corresponding to the peak ClNO$_2$ (Fig. 2h,p). Though gas-phase reaction Cl + NO$_2$ is discussed in the literature (Burkholder et al., 2015; Qiu et al., 2019a), however, to the best of our knowledge, such an unusually higher contribution of this reaction (seen in Delhi) as compared to the aqueous-phase reaction of Cl$^-$ + NO$_2^+$ has not been reported in any study. The reaction of Cl with NO$_2$ ($\approx 1.1 \times 10^5$ molec cm$^{-3}$ s$^{-1}$) is the major ClNO$_2$ source during sunrise in Leicester. In contrast, there is lesser contribution of Cl$^-$ + NO$_2^+$ reaction (rate $\approx 1 \times 10^3$ molec cm$^{-3}$ s$^{-1}$) in ClNO$_2$ production in Delhi. The prominent sink for ClNO$_2$ is through its heterogeneous reaction with Cl$^-$ ($\approx 1.8 \times 10^5$ molec cm$^{-3}$ s$^{-1}$ or $7.2 \times 10^{-15}$ mol mol$^{-1}$ s$^{-1}$) in Delhi almost throughout the day, while it's loss through the photolysis ($\approx 0.5 \times 10^5$ molec cm$^{-3}$ s$^{-1}$ or $2 \times 10^{-15}$ mol mol$^{-1}$ s$^{-1}$) is also an important sink during the daytime. We are using ClNO$_2$ uptake coefficient, $\gamma$ = 9E-3 from Fickert et al. (1998) in the simulation. Sensitivity simulation with $\gamma$ = 1E-5 (Haskins et al., 2019) results in considerably slower (by a factor of $\approx 270$ and $\approx 17$, near sunrise and during mid-day, respectively) loss rate of ClNO$_2$ with Cl$^-$ than in the NEW simulation over Delhi. ClNO$_2$ loss through the reaction ClNO$_2$ + Cl$^-$ ($\approx 2.7 \times 10^5$ molec cm$^{-3}$ s$^{-1}$ or $1.0 \times 10^{-14}$ mol mol$^{-1}$ s$^{-1}$) is its major sink in Leicester from mid-night to mid-day, while photolysis ($\approx 0.3 \times 10^5$ molec cm$^{-3}$ s$^{-1}$ or $1.1 \times 10^{-15}$ mol mol$^{-1}$ s$^{-1}$) is smaller sink from sunrise to mid-day here. The diurnal variation in Cl$_2$, and its production and loss mechanisms over Delhi and Leicester are shown by Fig. S1 and Fig. S2. In conjunction with major loss of ClNO$_2$, ClNO$_2$ + Cl$^-$ reaction is the major contributor to Cl$_2$ formation over Delhi and Leicester.

We also calculated ClNO$_2$ yield from NO$_2^+$ (Fig. S3), which is the ratio of P$_{ClNO_2}$/L$_{total}$, where P$_{ClNO_2}$ is the rate of ClNO$_2$ production through Cl$^-$ + NO$_2^+$ reaction and L$_{total}$ denotes the loss rate of NO$_2^+$ through it's reactions with Cl$^-$, H$_2$O, SO$_4^{2-}$, HCOO$^-$, CH$_3$COO$^-$, phenol, CH$_3$OH, and cresol (A4, A10–A16). ClNO$_2$ yield is $\approx 0.9$ over Delhi, indicating the strongest loss of NO$_2^+$ is through it's reaction with Cl$^-$, which is also mimicked in Fig. S4a showing the same concentrations of ClNO$_2$ as in NEW simulation and when other NO$_2^+$ reactions (A10–A16) are turned off (simulation: without other NO$_2^+$ reactions). ClNO$_2$ yield over Leicester is between $\approx 0.40$-$0.55$, which is about half the yield in Delhi. Stronger ClNO$_2$ yield in Delhi could be attributed to $\approx 2$ times higher Cl$^-$ than Leicester. Lesser ClNO$_2$ yield in Leicester portrays the importance of NO$_2^+$ loss reactions (A10–A15) other than with Cl$^-$, which could be seen through Fig. S4b where ClNO$_2$ is increased by more than twice during early morning hours when A10–A15 reactions are kept inactive in the model. The determination of ClNO$_2$ yield using cavity ring-down spectroscopy and chemical ionization mass spectrometry, shows yield ranging between 0.2 to 0.8 for Cl$^-$ concentrations of 0.02 to 0.5 mol/L (Roberts et al., 2009). The measurements of ClNO$_2$ yield for coastal and open ocean waters were found to be between 0.16-0.30 which is suppressed by up to 5 times than equivalent salt containing solutions, due to the addition of aromatic organic compounds (e.g., phenol and humic acid) to synthetic seawater matrices (Ryder et al., 2015).

## 4.3 Role of Cl in Atmospheric Oxidative Capacity (AOC)

In order to understand the role of Cl as oxidising agent with respect to the OH radical, we calculated the reactivity of Cl and OH as $\Sigma_{X_i}$ ($k_{radical+X_i} \times [X_i]$), where radical is Cl or OH, and [X$_i$] is the concentration of specie X$_i$ (here X$_i$ includes CO,

$CH_4$, primary VOCs and NMHCs which are initialized in the model) (Fig. 5). The corresponding rate constants for $Cl + X$ and $OH + X$ reactions are taken from the MECCA. The reactivity of both Cl and OH decreases rapidly nearly from sunrise to noon time and afterwards increases gradually at both locations. In comparison to Leicester, the magnitudes of Cl and OH reactivity in Delhi are higher by up to $\approx 1.4$ and $\approx 12$ times, respectively. However, the Cl/OH reactivity ratio in Leicester is up to $\approx 9$ times higher than that in Delhi. Cl reactivity is lower (Delhi: $\approx 685\,s^{-1}$, Leicester: $\approx 553\,s^{-1}$) during noontime and higher (Delhi: $\approx 750\,s^{-1}$, Leicester: $\approx 554\,s^{-1}$) during nighttime and early morning hours at both locations. The OH reactivity follows a similar pattern as that of Cl in Delhi and Leicester. The ratio of Cl to OH reactivity starts increasing after sunrise, reaching a maximum value of $\approx 42$ at nearly 16:00 h LT and then decreases further in Delhi. As mentioned above, Cl/OH reactivity ratio in Leicester shows a double peak pattern, with one peak ($\approx 270$) during early morning $\approx 04:00$ h LT and other peak ($\approx 276$) at about 16:00 h LT.

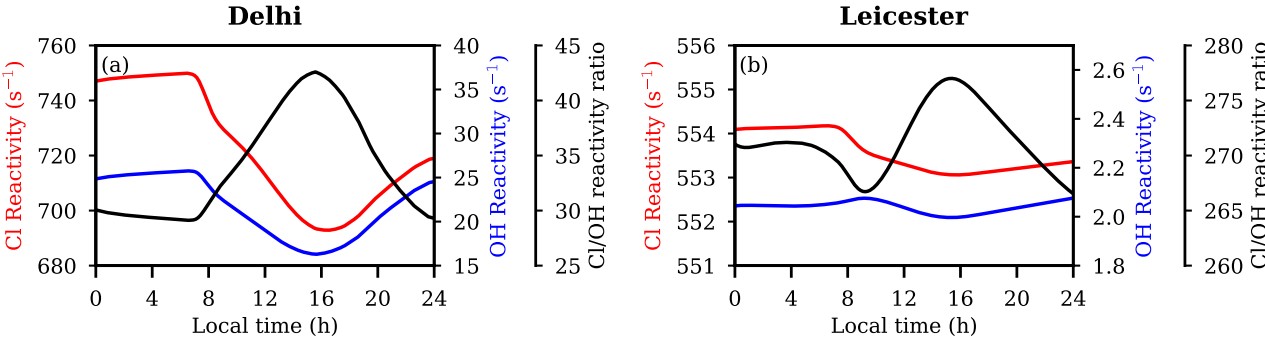

**Figure 5.** Reactivity of Cl and OH with $CO$, $CH_4$, and VOCs, and Cl/OH reactivity ratio during the simulation period in (a) Delhi and (b) Leicester.

We quantified the relative contribution of Cl in atmospheric oxidative capacity (AOC) using the model. AOC represents the sum of oxidation rates of specie $X_i$ by oxidants Y (OH, Cl, and other radicals: $NO_3$ and $O_3$) (Elshorbany et al., 2009):

$$AOC = \sum k_{X_i}\,[X_i][Y] \tag{1}$$

where, $k_{X_i}$ is the corresponding rate constant for $X_i$ + Y reaction. Accordingly, the magnitude of AOC depends upon the concentration and reactivity of Cl. Figure 6 shows the contribution of individual oxidants in AOC at both locations. Besides OH, Cl is the second most important oxidant in Delhi, with a significant contribution of 23.4 % during morning (averaged over 07:00-09:00 h LT), and 8.2 % throughout the day (06:00-16:00 h LT). In Leicester, Cl is the highest contributor (74.0 %) towards AOC during morning. In fact, with 34.1 % contribution, Cl is major oxidant after OH, during the daytime. Besides the abundance of Cl, higher reactivity enhances the contribution of Cl in AOC, which is further substantiated by the ratio of Cl reactivity to OH reactivity (Fig. 5b). This ratio indicates that Cl reactivity exceeds OH reactivity by a significant margin,

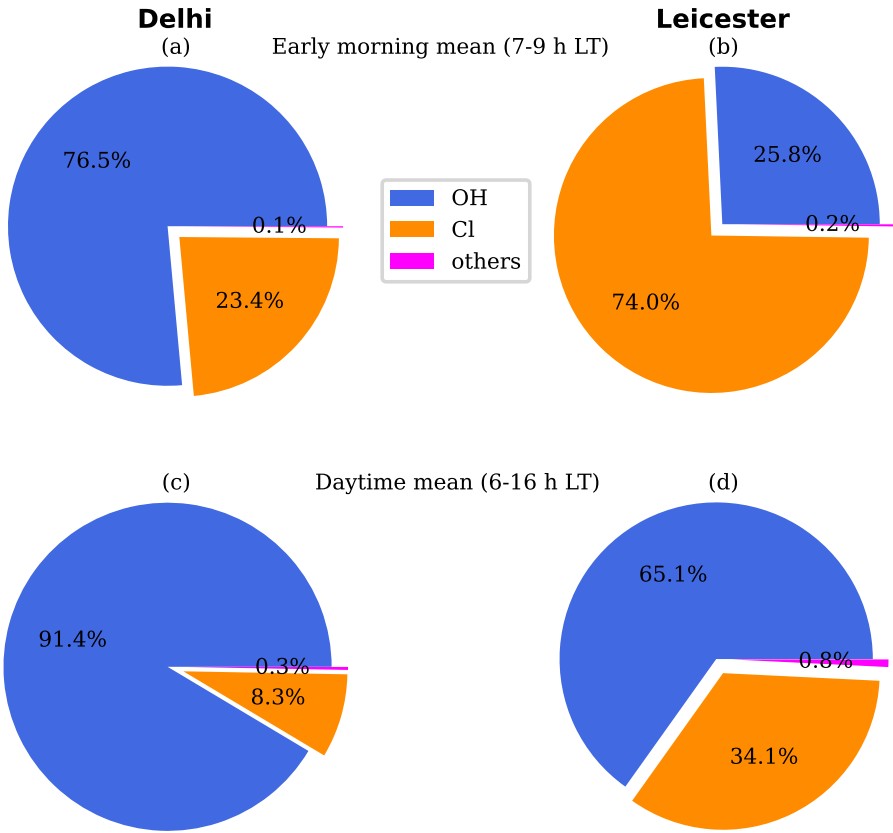

**Figure 6.** Atmospheric oxidative capacity (AOC) of radicals during (a, b) early morning mean (7-9 h LT) and (c, d) daytime mean (6-16 h LT) in Delhi (left panel) and Leicester (right panel).

ranging from 265 to 276 times greater throughout the day in Leicester. Such a substantial contribution of Cl in AOC leads to enhancements of $RO_2$ as seen in Fig. 3(f,l). Especially, a prominent peak in $RO_2$ during early morning (07:00-09:00 h LT) is imparted to strong participation of Cl in atmospheric oxidation during this time. Notably strongest contribution of Cl in AOC during early morning in Leicester, strengthens $RO_2$ peak by up to a factor of 8 (Fig. 3l). The role of Cl is predominant in Leicester as well as in Delhi during early morning, compared to a polluted environment of Hong Kong, China where Cl contribution was estimated to be 21.5 % (Xue et al., 2015). $NO_3$ and $O_3$ were found to play a relatively minor role in AOC at both urban environments.

### 4.4 Sensitivity to $ClNO_2$ + $Cl^-$ reaction

In a study conducted by Haskins et al. (2019), using the reacto-diffusive length-scale framework, it was demonstrated that field and laboratory observations could be reconciled by considering an aqueous-phase reaction rate constant for the $ClNO_2$ + $Cl^-$ reaction on the order of $\approx 10^4$ s$^{-1}$. This reaction rate constant is considerably lower (by $\approx$179 times) than reported in

Roberts et al. (2008). In this context, sensitivity simulation (NEWrate) is performed using a reaction rate coefficient of $5.6 \times 10^4$ $mol^{-1}$ $L$ $s^{-1}$ (Haskins et al., 2019) for the $ClNO_2 + Cl^-$ reaction, for both Delhi and Leicester. As depicted in Figure S7a, the concentration of Cl remains nearly the same in the NEWrate simulation compared to the NEW simulation over Delhi. However, there are significant changes in the concentration of $ClNO_2$, as shown in Fig. S7b. The simulated $ClNO_2$ exhibits a broader peak and is approximately 30 pmol/mol higher near sunrise in the NEWrate simulation when compared to the NEW

simulation. During the nighttime, approximately 20 pmol/mol of $ClNO_2$ is simulated in the NEWrate simulation, whereas it is negligible in the NEW simulation (see Fig. 3b). Since the Cl concentration is almost similar in both the NEW and NEWrate simulations, the differences in the simulated concentrations of OH, $HO_2$, and $RO_2$ remain consistent between the NEWrate or NEW simulations and the OLD and NOCL simulations (refer to Fig. S7d, e, f, and Fig. 3d, e, f). The production and loss mechanisms of Cl are similar in both the NEW and NEWrate simulations (see Fig. S8a and Fig. 4a). The contributions

from $ClNO_2$ formation reactions are also similar. However, in contrast to the NEW simulation, the loss of $ClNO_2$ through photolysis becomes dominant and is $\approx 6$ times greater than its loss through $ClNO_2 + Cl^-$ reaction, in NEWrate simulation. The contribution of radicals to AOC is also similar between the NEW and NEWrate simulation, as depicted in Fig. 6a,c and Fig. S9a,c respectively, over Delhi.

In contrast to Delhi, significant differences are seen in atmospheric composition in Leicester when the rate coefficient of the $ClNO_2 + Cl^-$ reaction is altered (as shown in Fig. S7). The peak concentration of Cl becomes $\approx 0.6$ fmol/mol during the morning hours of NEWrate simulation (Fig. S7g), which is about 4 times lower than the concentration of Cl in NEW simulation (Fig. 3g). However, due to slower rate of $ClNO_2$ consumption with $Cl^-$, the simulated $ClNO_2$ using the NEWrate is significantly enhanced (by $\approx 5$ times) compared to NEW simulation, reaching a maximum of about 210 pmol/mol around

sunrise (see Fig. S7h). Due to lower Cl concentrations, the levels of ClONO also decrease by 3.5 times in NEWrate simulation (as shown in Fig. S7i) compared to NEW simulation (Fig. 3i). The dominant peak seen at sunrise in the NEW simulation for OH, $HO_2$, and $RO_2$ is significantly reduced with the lower rate of the $ClNO_2 + Cl^-$ reaction, as illustrated in Fig S7j,k,l. Significant changes in the production and loss mechanisms of Cl and $ClNO_2$ are seen in Leicester when the reaction rate of A6 is changed, as shown in Fig. S8 and Fig. 4b. For example, in the NEWrate simulation, other reactions, including the

photolysis of $ClNO_2$ and ClONO, and $ClO + NO$ reaction, become prominent sources of Cl (with a rate of approximately 6.0 x $10^5$ molec $cm^{-3}$ $s^{-1}$), whereas in the NEW simulation, the major source for Cl is photolysis of $Cl_2$. The primary source for $ClNO_2$ production remains the $Cl^- + NO_2^+$ reaction in both the NEW and NEWrate simulations. However, in the NEWrate simulation, $ClNO_2$ loss from photolysis becomes the major sink, whereas in the NEW simulation, loss from the $ClNO_2 + Cl^-$ reaction is prominent. In addition, remarkable changes in AOC are seen between the NEWrate (Fig. S9b, d) and the

NEW simulation (Fig. 6b,d). In the NEWrate simulation, even though Cl remains the major oxidant its contribution is notably reduced from 74% (in NEW simulation) to 58.1% during the early morning hours.

## 5    Summary and Conclusions

Extended gas- and aqueous-phase chemistry of chlorine compounds has been added to the MECCA mechanism. It consists of 36 gas-phase reactions (inorganic, organic, and photolysis reactions). A total of 24 aqueous-phase and heterogeneous reactions have been added, containing detailed chemistry of $N_2O_5$ uptake on aerosols to yield $ClNO_2$ and various other competing reactions. The updated model is applied to two different urban environments: Delhi (India) and Leicester (United Kingdom) during winter time. The major conclusions are:

1. The model predicts up to 0.1 pmol/mol of $NO_3$ and up to 8 pmol/mol of $N_2O_5$ during daytime in Delhi. However, night-time production of $NO_3$ and $N_2O_5$ is seen to be negligible primarily due of the unavailability of $O_3$. In contrast to Delhi, $NO_3$ and $N_2O_5$ after mid-night in Leicester is $\approx$2.6 pmol/mol and $\approx$330 pmol/mol, respectively. $N_2O_5$ uptake on aerosols yields $ClNO_2$, which produces Cl via photolysis.

2. A sharp build-up of Cl with sunrise is mainly through $Cl_2$ photolysis in Delhi. Besides $Cl_2$, photolysis of $ClNO_2$ and ClONO and the reaction of ClO with NO are prominent Cl sources in Leicester. VOCs are the main sink for Cl at both locations, whereas $NO_2$ is also an important sink for Cl in Leicester. The latter results in the formation of $ClNO_2$ with a major contribution in Delhi, while $Cl^- + NO_2^+$ is a stronger source in Leicester. Photolysis is the major sink for $ClNO_2$ in Delhi, however, its uptake on chloride aerosols is a prominent sink in Leicester.

3. The magnitude of Cl ($\approx$750 s$^{-1}$) and OH ($\approx$25 s$^{-1}$) reactivities are significantly greater in Delhi, particularly during the morning hours, when compared to Leicester. However, Cl to OH reactivity ratio ($\approx$270) is pronounced in Leicester coinciding with higher contribution of Cl in AOC.

4. Sensitivity simulations reveal substantial post-sunrise enhancements of in OH, $HO_2$, and $RO_2$ radicals, with a prominent secondary peak due to Cl chemistry. Up to 8 times higher $RO_2$ is simulated in Leicester primarily because of leading role of Cl in AOC potential.

It is important to note that box models, despite their general limitation of neglecting transport phenomena and assuming species to be well mixed, do include highly detailed chemical mechanisms. Furthermore, because the model is initialized with measurements of chemical species at both locations and the modeled levels align with observed data, significant discrepancies in model estimates would be unexpected. Future studies focussing on modeling vertical gradients, in particular for radical reservoir species such as HONO, and $ClNO_2$ (Young et al., 2012) are recommended.

This study highlights the vital role of Cl chemistry in governing the oxidation capacity of the atmosphere and air quality, and therefore it is important to account for it in detailed photochemical as well as in 3-D chemical transport models. This will lead to better quantification of the importance of radicals in atmospheric oxidation and hence, the formation of ozone as well as secondary aerosols, over regional to global scale. Future studies focusing on secondary aerosol formation and new particle formation from heterogeneous reactions are needed to deepen the understanding of transformation of trace gases to aerosols.

*Code and data availability.* CAABA/MECCA is a community box model published under the GNU General Public Licence, available from the Gitlab repository (https://gitlab.com/RolfSander/caaba-mecca). The version of CAABA/MECCA updated in this study is currently available in the 'delhi' branch of the repository. The new chlorine mechanism will be included in the next release of CAABA/MECCA. All the model outputs associated with this study are archived at zenodo (https://zenodo.org/record/8332131; Soni et al. (2023)).

*Author contributions.* M. Soni, R. Sander, and D. Taraborrelli designed the study with inputs from S. S. Gunthe, P. Liu, and N. Ojha. M. Soni, R. Sander, and D. Taraborrelli developed and analyzed the chemical mechanism and M. Soni performed the simulations. A. Pozzer, R. Sander, L. K. Sahu, D. Taraborrelli, I. A. Girach, and N. Ojha helped M. Soni in the analyses and interpretations of the results. A. Patel assisted M. Soni in compiling literature and some input dataset. M. Soni wrote the manuscript and all the co-authors contributed to the review and editing.

*Competing interests.* At least one of the (co-)authors is a member of the editorial board of Atmospheric Chemistry and Physics.

*Acknowledgements.* The authors gratefully acknowledge the use of CAMS inventory for VOCs emissions data available from ECCAD (https://eccad3.sedoo.fr/catalogue). We thank ECMWF for the ERA5 dataset. We acknowledge UK AIR Air Information Resource for the chemical species data through https://uk-air.defra.gov.uk/data/. Authors thank J. M. Roberts (NOAA Chemical Sciences Laboratory, USA), Tao Wang (The Hong Kong Polytechnic University, Hong Kong), Men Xia (University of Helsinki, Finland), and Renuka Soni for valuable inputs on kinetics. M. Soni, N. Ojha, and L. K. Sahu acknowledge support from the Physical Research Laboratory, Ahmedabad, funded by the Department of Space, Government of India.

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
