# Peer review of "Comprehensive multiphase chlorine chemistry in the box model CAABA/MECCA: Implications to atmospheric oxidative capacity"

_EGUsphere, 2023_

## Author Comment (AC1)

**Comment:** Soni et al. report an updated mechanism for the community atmospheric chemistry box model CAABA/MECCA, adding some 35 gas-phase reactions, 15 aqueous-phase reactions, and 4 reactions governing the partitioning of gases between gas and aerosol phase. The focus is on chemistry of chlorine compounds ($Cl_2$, $ClNO_2$, etc.). Reactive intermediates such as the nitronium ion ($NO_2^+$) are treated explicitly. The revised model's was tested using two recent urban data sets collected in New Delhi (India) and Leicester (UK).

The paper is written well. I have a few concerns that the authors will hopefully be able to address in revision.

**Response: We thank the reviewer for the comments and suggestions, which have improved our manuscript. Please find our responses to the comments below in blue fonts. The discussion added/updated in the manuscript is presented by red color font.**

**General comments.**

**1.** More explanation is needed to justify the chlorine nitrite chemistry in the mechanism, considering this molecule has not been unambiguously observed in ambient air. The authors are correct that chlorine nitrite may form from $Cl+NO_2$ (e.g., Golden, J. Phys. Chem. A 2007, 111(29), 6772–6780, https://doi.org/10.1021/jp069000x and Niki et al., Chem. Phys. Lett. 1978, 59(1), 78-79, https://doi.org/10.1016/0009-2614(78)85618-8) - these papers should be cited. However, its chemistry is incomplete. ClONO is metastable and converts to $ClNO_2$ (Janowski et al., Berichte der Bunsengesellschaft für physikalische Chemie 1977, 81(12), 1262-1270, https://doi.org/10.1002/bbpc.19770811212; Niki et al. Chem. Phys. Lett. 1978, 59(1), 78-79, https://doi.org/10.1016/0009-2614(78)85618-8) - a reaction that should be added to the mechanism. Note that Niki et al. report a ClONO lifetime of ~150 s.

**Response: We thank the reviewer for pointing out the important references to improve the understanding of ClONO to $ClNO_2$ mechanism. We have incorporated the suggested reaction of the metastable state ClONO converting it into $ClNO_2$ as proposed by Janowski et al., 1977. According to their work, the conversion time between ClONO and $ClNO_2$ ranges from 4 to 20 hours. In our manuscript, we have adopted a conversion time of 12 hours (average of 4 and 20 hours) and consequently, the corresponding rate constant is calculated to be 2.3E-5 $s^{-1}$. This discussion is added in the manuscript citing the suggested references.**

**Lines 83-86 : "ClONO is formed through reaction of Cl with $NO_2$ (G2), and exists as a metastable intermediate (Janowski et al., 1977, Niki et al., 1978, Golden, 2007). This intermediate subsequently transforms into $ClNO_2$ (G10), with an average conversion time of ≈12 h (ranging from 4 to 20 h), and the corresponding rate constant is 2.3 E-5 $s^{-1}$ (Janowski et al., 1977)."**

**Table 1: "(G10) ClONO → $ClNO_2$; 2.3E-5 $s^{-1}$; Janowski et al. (1977)"**

Be it as it may, the reaction between Cl and $NO_2$ is generally thought to be negligible compared to reaction of $N_2O_5$ on chloride containing aerosol, except for unusual environments such as Delhi in winter. This paper thus seems to be tailored towards specific data sets, which should be mentioned in the introduction, and the relevant measurement papers (e.g., Haslett et al., Atmos. Chem. Phys. Disc., https://egusphere.copernicus.org/preprints/2023/egusphere-2023-497/) should be cited.

**Response: Yes, generally Cl + NO$_2$ reaction is negligible or much smaller compared to N$_2$O$_5$ reaction on aerosols. However, inclusion of this chemistry makes the model more comprehensive and our results highlight the implications of such different reactions in two distinct urban environments (Delhi as well as Leicester). For example, gas phase reaction of Cl with NO$_2$ contributes significantly to the ClNO$_2$ formation in Delhi, while aqueous phase reaction of Cl$^-$ + NO$_2^+$ is dominant in Leicester. Therefore, incorporating all these reactions into the model is essential for better and more complete understanding of the Cl chemistry. As recommended by the reviewer, we have included a discussion of the relevant paper (Haslett et al., 2023) addressing the unusual chemistry observed in Delhi in the revised version of the manuscript.**

**Lines 177-179: "Similar unusual daytime high levels of N$_2$O$_5$ ($\approx$21.9±29.3 pptv) during wintertime were recently measured over Delhi using a high-resolution iodide adduct chemical ionization mass spectrometer (Haslett et al., 2023)"**

**2.** More discussion as to the applicability of this model is needed.

For example, a limitation of this study is that all species are assumed to be well-mixed. In reality, there will be vertical gradients for most species evaluated here, in particular radical reservoir species such as HONO and ClNO2 (e.g., Young et al., Environm. Sci. Technol. 2012, 46(20), 10965-10973, https://doi.org/10.1021/es302206a). This limitation should be discussed.

**Response: As suggested, we have added discussions in the applicability of this model including the suggested references. The discussion regarding the generic limitations of the box model (vertical gradients, transport, etc.) and recommendation for future study is mentioned in toward the end of manuscript.**

**Lines 361-365: "It is important to note that box models, despite their general limitation of neglecting transport phenomena and assuming species to be well mixed, do include highly detailed chemical mechanisms. Furthermore, because the model is initialized with measurements of chemical species at both locations and the modeled levels align with observed data, significant discrepancies in model estimates would be unexpected. Future studies focussing on modeling vertical gradients, in particular for radical reservoir species such as HONO, and ClNO$_2$ (Young et al., 2012) are recommended."**

Transport phenomena should also be acknowledged (since the model does not include them) and assumptions should be clearly stated. For example, it is well established that ClNO2 formation occurs in the nocturnal residual layer (which contains less NO than the surface layer), and ClNO2 then mixes downward in the morning when the convective mixed layer forms (e.g., Bannan et al., J. Geophys. Res. 2015, 120(11), 5638-5657, https://doi.org/10.1002/2014jd022629; Tham et al. Atmos. Chem. Phys. 2016, 16(23), 14959-14977, https://doi.org/10.5194/acp-16-14959-2016).

**Response: We agree with the reviewer that absence of transport phenomena would cause deviations in variability of species when compared with actual observations. In the revised version of the manuscript we have acknowledged the previous studies (Bannan et al., 2015, Tham et al., 2016) showing the formation of ClNO$_2$ in the nocturnal residual layer, which contains lower levels of ClNO$_2$ and then it mixes downward during the morning when mixed**

**layer forms. We have also added the limitation of box model which does not include the effects of transport. The  following discussion is added to the manuscript:**

**Lines 219-222: "Previous studies have demonstrated that the formation of ClNO$_2$ occurs within the nocturnal residual layer, which contains lower levels of NO compared to the surface layer. Subsequently, ClNO$_2$ mixes downward during the morning when the convective mixed layer develops (Bannan et al., 2015; Tham et al., 2016). However, the present study does not account the the effect of transport processes due to the limitations of the box model."**

**Specific comments**

**1.** Title - specify season of study in title of paper (winter)

**Response: Although we have presented model results from a newly developed chlorine chemistry mechanism for the winter season, this mechanism also holds during other seasons, possibly improving the estimation of the oxidation capacity of the atmosphere across various photochemical states. In light of this perspective, our title, without specifying the season of the study, would effectively emphasize the wide-ranging implications of the developed chlorine chemistry module. Therefore, we wish to retain the title of the manuscript.**

**2.** line 27 - The authors differentiate between nitryl chloride (ClNO$_2$) and chlorine nitrite (ClONO), citing a the MCM modeling study by Riedel et al (2014). More explanation is needed here since Riedel et al. do not mention chlorine nitrite in their paper.

**Response: Thanks for pointing this out. We have added the correct reference Atkinson et al., 2007 which shows that photolysis of ClONO produce Cl radicals (Line: 28).**

Chlorine nitrite may form from Cl+NO$_2$ (e.g., Golden, J. Phys. Chem. A 2007, 111, 29, 6772–6780, https://doi.org/10.1021/jp069000x and Niki et al., Chemical Physics Letters 1978, 59(1), 78-79, https://doi.org/10.1016/0009-2614(78)85618-8); however, the reaction between Cl and NO$_2$ is generally thought to be negligible compared to reaction of N$_2$O$_5$ on chloride containing aerosol. It would thus be informative to add the relative contributions of ClONO and ClNO$_2$ to the bottom trace of Figure 2.

**Response: As suggested by the reviewer, the individual concentrations of ClNO$_2$ and ClONO are now shown in Figs. 3b, h and 3c, i, respectively.**

**3.** line 3. Please define the abbreviation CAABA/MECCA.

**Response: It is already defined, in the first sentence in mechanism development section.**

**4.** line 80. Please state here that the full mechanism is shown in the SI.

**Response: Now we have mentioned in the revised manuscript (Lines: 77-78).**

**5.** Page 4 - Table 1. Please state the units for the reaction rate constants.

**Response: The rate constants mentioned in Table 1 are mostly in units of $cm^3$ molecule$^{-1}$ s$^{-1}$, which is now mentioned in the table's caption. Unit of reaction G10 and photolysis reactions (s$^{-1}$) is mentioned in the table. The updated table caption:**

**"Gas-phase chlorine reactions and corresponding rate constants added to MECCA. The rate constants are expressed in units of $cm^3$ molecule$^{-1}$ s$^{-1}$ unless otherwise specified. Model-simulated maximum noontime J-values for Delhi are provided."**

**6.** Page 4 - Table 1. For the photolysis reactions, please state the maximum (noon) j values.

**Response: Model simulated maximum J-values for Delhi are now added to the Table 1, and table's caption is updated accordingly.**

**7.** Page 6, Table 2 - reaction A13 - please subscript the 3 in CH3COO.

**Response: The typographical error is now corrected in the revised manuscript.**

**8.** Line 100 - The Sander et al. (2014) reference is inappropriate. Cite Ghosh et al., J. Phys. Chem. A 2012, 116, 5796-5805, https://doi.org/10.1021/jp207389y, please.

**Response: We agree that Ghosh et al. 2012 should be added as the reference for the UV/VIS absorption spectrum of nitryl chloride. The paper by Sander et al. (2014) is cited because it explains how cross sections are converted to photolysis rate constants (j-values) in the model. Now, Ghosh et al., 2012 is also cited along with Sander et al., 2014 in the revised manuscript (Line: 104).**

**9.** pg 7 - Figure 1 - The nitronium ion ($NO_2^+$) is a potent nitrating agent, and there are many more organic molecules in the aerosol-phase than shown here. Please discuss the limitations of the abridged mechanism.

**Response: We thank referee to emphasize on this point. We agree that $NO_2^+$ is a potent nitrating agent which could participate in with many more aqueous phase reactions with organic molecules than shown here. However, the chemical kinetics for nitration reactions of other alcohols (e.g. catechol and polyphenols) are not available in literature, and the rate constant for nitration of methoxyphenol is ~10000 times smaller than nitration of phenol. Coombes et al., 1979 reports that rate constant for nitration of phenol and cresol is similar, based on which $NO_2^+$ reaction of cresol is also added to the model and simulations are updated accordingly. The manuscript is updated as follows:**

**Figure 1, Table 2 are updated and above discussion is added to the revised manuscript as:**

**Lines 111-112: " ..... cresol (A11-A16) (Staudt et al., 2019; Ryder et al., 2015; Heal et al., 2007; Iraci et al., 2007; Coombes et al., 1979)."**

Lines 116-120: "The rate constant for $NO_2^+$ reaction with methoxyphenol is about $\approx 10000$ times smaller than $NO_2^+$ + phenol reaction (Kroflič et al., 2015), so it is not considered in this study. In addition, nitration reactions of other alcohols (e.g. catechol and polyphenols) could be potentially important, however due to unavailability of corresponding rate constants, these reactions are not considered in this study, nonetheless future studies calculating the kinetics of these reactions are recommended."

**10.** pg 10 - Figure 2: It would be informative to add the relative contributions of ClONO and $ClNO_2$ to the bottom trace of Figure 2.

**Response: The concentrations of ClONO and $ClNO_2$ are now added separately to the figure 3.**

pg 10 - Figure 2: The $NO_3$ and $N_2O_5$ peaks at noon local time are highly unusual. Consider adding a brief explanatory note to the caption.

**Response: The diurnal pattern of $NO_3$ and $N_2O_5$ is really unusual in Delhi. The main reasons for negligible $NO_3$ at night is because of zero $O_3$, which is due to its titration by the high NO concentrations. Although mixing ratios of $NO_3$ and $N_2O_5$ peak during daytime, their levels remain quite low. As suggested by reviewer, following brief explanatory note is added regarding the noticable unusual levels to the figure's caption:**

**".......The unusal and negligible nighttime $NO_3$ in Delhi is attributed to the nearly non-existent $O_3$, due to titration by higher concentrations of NO. This leads to the negligible nighttime $N_2O_5$ in this region. Although mixing ratios of $NO_3$ and $N_2O_5$ peak during the daytime, their levels remain quite low....."**

**11.** pg 10 - Figure 3. The left-hand side graphs suggest that there is no nitryl chloride formation from heterogeneous uptake of $N_2O_5$ in Delhi. This seems unlikely considering non-zero mixing ratios of $N_2O_5$ are shown in Figure 2 (see also the next comment).

**Response: $P(ClNO_2)$ is small but not zero, it is not visible in the plot due to large scale of y-axis. However, the plot on log-scale depicts its production through $Cl^- + NO_2^+$ in order's of about $10^3$ molec cm$^{-3}$ s$^{-1}$. This point is clarified in the revised manuscript.**

Lines 251-252: "In contrast, there is lesser contribution of $Cl^- + NO_2^+$ reaction (rate $\approx 1 \times 10^3$ molec cm$^{-3}$ s$^{-1}$) in $ClNO_2$ production in Delhi."

**12.** pg 10 line 201. Morning $ClNO_2$ peaks are generally due to vertical transport of $ClNO_2$ produced in the residual layer to the surface. Please cite relevant literature here and discuss.

**Response: The relevant literature is discussed in following lines in the manuscript (similar to response of general comment 2; Lines: 219-222).**

---

## Author Comment (AC2)

**General Comments**

Soni et al present new model results for air quality simulations that include the impacts of chlorine chemistry. The manuscript reports on updates to the chemical mechanism of CAABA/MECCA and discusses the impact of chlorine chemistry in two disparate regions (Leicester and Delhi). I have several general and specific comments that should be addressed prior to publication.

**Response: Thank you for the constructive review; responding to these comments has improved our manuscript. Please find our responses below in blue fonts. The discussion added/updated in the manuscript is presented by red color font.**

- The model reports a surprisingly large conversion of ClNO2 to Cl2. This is because of the large, condensed phase rate constant for ClNO2 + Cl- that was implemented in the model (Roberts et al 2008). More recent analyses have shown that this rate is likely too large. For example, the analysis of Haskins et al., JGR 2019, using field observations of ClNO2 and Cl2 suggested that this rate must be significantly smaller (of order 1E4 s-1). I think the authors should look at a sensitivity test to this rate to highlight that the selection of the ClNO2+Cl- rate constant has a really significant impact on the Cl production rate.

    **Response: We thank the reviewer for raising this point and suggesting the article by Haskins et al., 2019. We agree that the aqueous phase rate constant of $ClNO_2 + Cl^-$ reaction has significant effect on the production rate of Cl radicals as well as $ClNO_2$. As a consequence of including this reaction, the contribution of various reactions participating in Cl and $ClNO_2$ formation should change. In addition, significant changes could occur in concentrations of OH, $HO_2$, and $RO_2$ radicals in NEW simulation which contains the newly added Cl chemistry. These changes would also reflect in AOC over both the locations. In this regard, as suggested by reviewer a sensitivity test is performed with reaction rate of 5.7E4 $mol^{-1}$ L $s^{-1}$ and its effects are discussed by introducing a new section (4.4) in the revised manuscript. Figures S7 (simulated diurnal variations in Cl, $ClNO_2$, ClONO, OH, $HO_2$, and $RO_2$), S8 (Cl and $ClNO_2$ budget), and S9 (contribution of radicals in AOC) are added to the supplement to depict the changes occurred due to reaction rate of $ClNO_2 + Cl^-$. The above discussion is added in new section 4.4, and supplementary figures S7-9.**

- As I understand the model treatment of heterogeneous and multiphase reactions here is quite different than what is in most models. Specifically, it appears that N2O5 is equilibrated between the gas and condensed phase using an equilibrium constant then permitted to react. It would be very helpful if the authors compared (and contrasted) this approach to the more common approach of using a reactive uptake coefficient for N2O5 chemistry that is sensitive to the chemical composition and phase of the aerosol particles. I expect that the two approaches would yield quite different results both with respect to magnitude and temporal trends. Since N2O5 chemistry is central to this study, this should be discussed.

    **Response: In principle, employing a common approach for $N_2O_5$ chemistry using reactive uptake coefficients depending on chemical composition is expected to yield similar results, provided that the implementation of gammas and numerical integration is done properly. This approach essentially includes the rate coefficients from the fully**

explicit kinetic model. In other words, the more common approach is a simple parameterization of the detailed aqueous-phase $NO_2^+$ chemistry that we have in our model and presented in the manuscript. It is worth noting that our model has the explicit reactions (forward and backward) for phase partitioning which often results in an effective phase partitioning equilibrium being established, consequently our approach is better compared to conventional approach. We do expect similar results with the two approaches but we think that such an analysis is beyond the scope of the present manuscript.

- Since the model has a comprehensive gas phase chemical mechanism, and the authors draw conclusions about the role of OH vs Cl radicals in VOC oxidation, it would be a nice opportunity to comment on the production of oVOC that stem directly (and only) from Cl+VOC reactions as they could be used in the future for testing the role of Cl chemistry. For example, what is the mixing ratio of chloroacetone?

  Response: Cl reacts with hydrocarbons and acetone via H-abstraction. This does not form any Cl-containing molecules like chloroacetone. We don't have any such reactions in MECCA where the attacking Cl atom remains inside the organic molecule. In a future study, it could be checked if there are any important additions of Cl to a double bond which could form Cl-containing molecules. This discussion is now reflected in revised manuscript as outlook towards the end:

  Lines 120-123: "In this study, Cl reacts with hydrocarbons and acetone via H-abstraction, and hence does not lead to the formation of any Cl-containing molecules, such as chloroacetone. This means that there are no such reactions in MECCA in which the Cl atom becomes part of the organic molecule. Therefore, for future research, it would be valuable to investigate the chemical kinetics of such reactions kinetics and their importance in the formation of organohalogen compounds."

**Specific Comments**

**1.** Line 55: This reaction is listed as H1, but H2 (and so on) are all Henry's Law Equilibriums. Should this be R1?

Response: H was currently used both for Henry's law equilibrium reactions and for heterogeneous non-equilibrium reactions. To avoid any confusions, we are now using "HET1" instead of "H1" in the revised manuscript (Lines 55-56).

**2.** Line 56: I would remove "recent studies" as it was shown 1997 by Behnke et al that the heterogeneous reaction of N2O5 could form ClNO2 on aqueous chloride containing films.

Response: We agree with the reviewer. The "recent studies" is removed and the sentence is updated as:
Line 57: "However, $N_2O_5$ uptake on chloride-containing particles can produce $ClNO_2$ (Behnke et al., 1997; Thornton et al., 2010), ..."

**3.** Table 2: It would be helpful to add the Henry's Law constants (and references) to the table for the molecules studied. The Henry's Law constant for most of these gases have never been measured, thus the values reported in the literature are based on model fits to measured reactive uptake coefficients.

**Response: We agree with the reviewer. The Henry's law constants with references are now added to Table 2.**

**4.** Section 3: How is aerosol surface area and aerosol liquid water treated in the model? I appreciate that this may be described in one of the cited references (Rosanka?) But given its central importance to the science discussed here, I think it would be helpful if there was an explicit discussion.

**Response: The following aerosol properties are defined in the model: radius, liquid water content, and chemical composition for both Delhi and Leicester, which are also mentioned in supplementary table S1. A line is added to the revised manuscript for clarification:**

**Line 151-153: "The values of aerosol properties (e.g., radius, liquid water content, and chemical composition) incorporated in the simulations for both Delhi and Leicester as provided in Table. S1."**

**5.** Line 203: I think it would be helpful to cast the ClNO2 loss rates in units of per second as it is easier for readers to compare them to other locations.

**Response: The ClNO$_2$ loss rates are defined in units of per second (molec cm$^{-3}$ s$^{-1}$) in the manuscript. In addition, we have now also added the ClNO$_2$ loss rates in mol mol$^{-1}$ s$^{-1}$ in the text (Lines 253-257).**

**6.** The ClNO2 + Cl- loss rate is enormous, and it would be helpful to see how that compares to other locations. Specifically, the ClNO2 uptake coefficient is quite small (< 1E-5) even on acidic aerosol. The surface area here must be enormous to drive a loss rate that is 10x that of photolysis (3hr lifetime). I think it would be very helpful to expand the discussion here to think more directly about this comparison.

**Response: We are using alpha = 9.0E-3 from Fickert (doi:10.1021/JP983004N), not <1E-5.**

---

## Author Comment (AC3)

**Comment:** The manuscript "Comprehensive multiphase chlorine chemistry in the box model CAABA/MECCA: Implications to atmospheric oxidative capacity" by Soni et al. describes an expansion of the MECCA chemical mechanism to include chlorine chemistry. Using published data from India and the UK, the authors show how the inclusion of this additional chemistry leads to improved modelling results and present an interesting analysis of the oxidation chemistry in these two very different locations.

The manuscript is generally well written, although the English could benefit by some tweaking, and clearly laid out. I have only a few comments, and after the authors have addressed them, I recommend publication.

**Response: We thank reviewer for the constructive comments to our manuscript. Please find our responses below in blue fonts. The discussion added/updated in the manuscript is presented by red color font.**

**1.** My main suggestion is to change figures 2 and 6. I think it would make the whole paper much clearer if they both show the base model, the base model with added chemistry, and the measurements. To keep the figures in a manageable size I would suggest having all radical species in one figure and all non-radicals species in the other figure. Likewise, I suggest introducing earlier in the paper the three mechanisms that are now discussed only from section 4.3 onwards. In this way, it will be easier for the reader to understand how the model results have changed with the addition of the new Cl chemistry.

**Response: As suggested, we have modified Figure 2 and 6 (which is now Figure 2, 3 in the revised manuscript). To manage the size of the figures and lay out the discussion clearly, we have moved Cl, ClNO$_2$, and ClONO to Figure 3 of the revised manuscript. In our view, introducing three simulations from Figure 2 or at the beginning of Section 4 would make the discussion a bit chaotic. Inf act, as the concentration of NO and NO$_2$ is constrained in the model simulations, the diurnal levels of NO, NO$_2$, and O$_3$ that are simulated by the three model runs will coincide with each other. As a result, noticable changes in the diurnal levels of NO$_3$ (which forms through the reaction of NO$_2$ + O$_3$) and N$_2$O$_5$ cant't be seen when the three model runs are shown together. Therefore, for better manuscript flow, we have defined the three simulations in Section 4.1 of the manuscript. We have also modified the names of the model runs in order to avoid any confusion, and in the revised version, the model runs are referred to as follows:**

**OLD=includes default chemistry already present in the model**
**NEW=chemistry already present in the model + newly added gas and aqueous phase chlorine chemistry**
**NOCL=OLD minus chlorine chemistry (i.e. without Cl chemistry).**

**The following line is added to clarify that OLD simulation also include some basic chlorine chemistry that was already present in MECCA before we started model development.**
Lines 192-194: "OLD simulation also encompassed some basic chlorine chemistry that was part of the model prior to its update (full mechanism is also shown in supplement)."

**2.** line 36: I wouldn't say that the limitation in our understanding of Cl chemistry is "mostly" due to the limitations of the models. These processes are also understudied in laboratory/chamber experiments, not to mention that the database of ambient observations is rather limited.

**Response: As suggested, the following line is added to reflect that chemistry of Cl compounds are understudied in laboratory/chamber experiments.**

**Line 39: "In addition, the chemistry of Cl compounds has been less studied using the laboratory/chamber experiments."**

**3.** lines 125-127. I suggest moving to line 121 the explanation of why the winter season was chosen for the model simulations, and also add a note explaining why the Leicester and Delhi datasets were used for this study.

**Response: The motivation of choosing winter season for model simulation is now moved as suggested (Lines 132-134 of revised manuscript).**

**4.** figure 2: the isoprene mixing ratio in Leicester looks constant. I assume it is an estimate of some sort, and in an average sense that may be fine, but the profile is likely unrealistic. The authors should consider how this affect their results and the related discussion.

**Response: The constant value shown in Figure 2 represents the observations, not the model. This is already mentioned in line 162 of the revised manuscript and is now also clarified in the Figure 2 caption. A diurnal cycle of measured isoprene is not available for Leicester, and therefore, the mean value is used to illustrate that the modeled isoprene varies around the observed mean level.**

**5.** line 211: "indicating", rather than "representing"?

**Response: In the revised manuscript, "representing" is now replaced by "indicating".**

**6.** line 219: "Cl- concentrations"?

**Response: Yes, we have updated as suggested.**

**7.** line 226: why are the rate constants for OH + X reactions not taken from MECCA, like those for Cl + X reactions?

**Response: The rate constants for nearly all the OH+X reactions were already published in Soni et al., 2022, and those were based on another box model, NCAR's master mechanism. Hence, they were directly taken from that reference. However, as correctly pointed out by the reviewer, the rate constants do vary in different models. Therefore, in the revised manuscript, all the rate constants are taken from MECCA only, and the calculations are revised accordingly (Line 279, Figure 5).**

**8.** figure 3, and related discussion: the model suggests that the gas phase reaction Cl + NO2 can be a significant source of ClNO2. As far as I am aware, most studies indicate the aqueous-phase reaction as the major (if not only) source of ClNO2, so this may be a potentially interesting/important finding. Can the authors expand the discussion on this point? For instance, how well is this reaction known? Have previous studies considered it?

**Response: We agree with the reviewer, and similarly, reviewer #1 also pointed out that the contribution from the gas-phase reaction $Cl+NO_2$ is thought to be negligible compared to the aqueous-phase reaction of $Cl^- + NO_2^+$ in the formation of $ClNO_2$. The chemistry presented over the Delhi environment is quite unusual during wintertime, such as the nighttime negligible and daytime peak levels of $NO_3$ and $N_2O_5$. Measurements of such an unusual diurnal pattern of $N_2O_5$ are also reported in a recent study by Haslett et al., 2023 (which is discussed in the revised manuscript, Lines: 177-179). Though gas-phase reaction $Cl + NO_2$ is discussed in the literature (Burkholder et al. 2015, Qiu et al., 2019), however, to the best of our knowledge, such an unusually higher contribution of the gas-phase $Cl+NO_2$ reaction as compared to the aqueous-phase reaction of $Cl^-+NO_2^+$ has not been reported in any study (discussed in the revised manuscript, Lines: 247-250). In fact, the detailed budget of $ClNO_2$ considering a comprehensive set of gas and aqueous-phase reactions of involved reactions along with showing the importance of different production and loss mechanisms of $ClNO_2$ in distinct urban environments are not presented anywhere in the literature. In this regard, our results provide more comprehensive insights and highlight the implications of these different reactions in urban environments.**

**9.** figure 4, and related discussion: I find it a bit odd that Cl is so important for the AOC in Leicester when the model predicts significant concentrations of Cl only around 8am. Likewise the levels of Cl in Delhi during the night are expected to be very small. Perhaps the authors should comment on this point.

**Response: (Considering the reviewer is pointing towards Figure 5 (showing AOC) and related discussion)**

**As reviewer pointed, it is correct that the model predicts significant concentrations of Cl at around 8 am over Leicester. Since morning time (7-9 h LT) strong contribution is included in the mean value of AOC during daytime (6-16 h LT), higher Cl reactivity throughout the day lead to stronger contribution from Cl in daytime (6-16 h LT) AOC in Leicester. In addition, results reveal a significant change in AOC in Leicester with the changes in reaction rate coefficient of $ClNO_2 + Cl^-$ reaction. For example, morning-time AOC dropped from 74% to 58.1%. A new section (4.4) has been added to the manuscript discussing the changes occurring due to the reaction rate coefficient of the A6 reaction.**

**As per reviewer's second point, "Likewise the levels of Cl in Delhi during the night are expected to be very small", Cl concentration is zero during the night as expected which is clearly seen in Figure 3a.**

**10.** lines 251-257: it is not clear to me how the base model differ from the base model without chlorine chemistry. Up until this point I was under the impression that chlorine chemistry was not

present in the "original" MECCA. Can you please clarify here, and in the Introduction if necessary, what are the differences in the various mechanisms?

**Response: Some basic chlorine reactions were already included in MECCA before we initiated this work (full mechanism is included in supplement). To prevent any confusion, we have modified the simulation names as discussed in response to comment (1). Additionally, we have added following line for further clarification.**

**Line 192-194 : "OLD simulation also encompassed some basic chlorine chemistry that was part of the model prior to its update (full mechanism is also shown in supplement)."**

---

## Author Response (AR3)

**General comment:** The authors provide an interesting box-model study showing the impacts of extended Cl chemistry on two distinct urban environments. While the majority of the reviewers' comments have been addressed in the revised versions, I have a few remaining questions, and also some minor typographical errors (listed in the 'private note' to authors) that should be addressed. I noted a few points in going through the response to reviewers and the manuscript that could be improved.

**Response: We are grateful for a careful evaluation of the revised manuscript and valuable comments. Our responses are given below in blue-color fonts and corresponding changes in the manuscript are highlighted in red-color.**

**Comment 1:** Reviewer 2 noted the possibility of the formation of organohalogens. I see that Cl+isoprene (which is known to make organohalogens) is included in the model, but (presumably for simplicity?) the chemistry in the model mimics the OH reaction. Maybe a brief statement could be made along these lines?

**Response: Yes, a simple mechanism mimicking the OH + isoprene has been included for Cl + isoprene reaction. Developing a kinetic model for the isoprene oxidation forming organohalogens could be another major project worth addressing in a future study. As suggested, we have clarified this aspect in the revised manuscript, as follows:**

**Lines 122-124: "The reaction of Cl atoms with isoprene proceeds mainly via addition, and it produces chlorine-containing organics (Ragains and Finlayson-Pitts, 1997; Fan and Zhang, 2004). However, here we have simplified the mechanism by not considering the fate of organohalogens."**

**2.** Line 286, perhaps elsewhere. The fact that Cl-reactivity is high compared to OH reactivity does not necessarily mean that Cl contributes significantly to oxidizing capacity (there need of course to be Cl-atoms present). Please clarify.

**Response: We agree. Apart from higher Cl reactivity compared to OH, abundance of Cl-atoms should be significant to influence the oxidation capacity. The discussion has been revised suitably, as follows:**

**Lines 297-298: Accordingly, the magnitude of AOC depends upon the concentration and reactivity of Cl.**

**Lines 301-303: Besides the abundance of Cl, higher reactivity enhances the contribution of Cl in AOC, which is further substantiated by the ratio of Cl reactivity to OH reactivity (Fig. 5b).**

**Lines 363-364: However, Cl to OH reactivity ratio (≈270) is pronounced in Leicester coinciding with higher contribution of Cl in AOC.**

**3.** Please add anything further to the specific comment #6 of R2 regarding the ClNO2 uptake coefficient, if possible.

**Response: We performed a sensitivity simulation with $ClNO_2$ uptake coefficient of 1E-5 and compared the loss rate of $ClNO_2$ with $Cl^-$ in this simulation with NEW simulation over Delhi.**

**Accordingly, the following discussion has been added in the revised manuscript to mention the role of ClNO$_2$ uptake coefficient.**

**Lines 256-259: We are using ClNO$_2$ uptake coefficient, γ = 9E-3 from Fickert et al. (1998) in the simulation. Sensitivity simulation with γ = 1E-5 (Haskins et al., 2019) results in considerably slower (by a factor of ≈270 and ≈17, near sunrise and during mid-day, respectively) loss rate of ClNO$_2$ with Cl$^-$ than in the NEW simulation over Delhi.**

**Minor 'typos':** Please replace the current text with that shown below:
**Response: All the typos pointed out by the editor, listed below, have been corrected in the revised manuscript.**

Line 14 (and elsewhere): 'near sunrise'.

Line 17: the atmosphere and radical cycling...'

Line 32: 'associated mechanisms'

Line 33: delete 'however'

Line 45: "A few recent studies..."

Line 115: 'The heterogeneous chemistry just discussed is implemented..."

Line 182: "gets removed rapidly during..."

Line 228-9: "due to Cl chemistry, further measurements..."

Line 232: "left-upper panel (a)"

Line 236: "to the morning peak, which have ..."

Line 252: its (not it's)

Line 369: "to better quantification of the importance..."